# Research

ecology, palaeontology

stable isotope analysis, ecology, niche, herbivore

**Author for correspondence:**
Melissa I. Pardi
e-mail: melissa.pardi@illinois.gov

# Dietary plasticity of North American herbivores: a synthesis of stable isotope data over the past 7 million years

Melissa I. Pardi[1,2] and Larisa R. G. DeSantis[1,2]

[1]Department of Biological Sciences, and [2]Department of Earth and Environmental Sciences, Vanderbilt University, Nashville, TN 37235, USA

MIP, 0000-0001-9766-7511; LRGD, 0000-0003-1159-9154

Palaeoecological interpretations are based on our understanding of dietary and habitat preferences of fossil taxa. While morphology provides approximations of diets, stable isotope proxies provide insights into the realized diets of animals. We present a synthesis of the isotopic ecologies ($\delta^{13}$C from tooth enamel) of North American mammalian herbivores since approximately 7 Ma. We ask: (i) do morphological interpretations of dietary behaviour agree with stable isotope proxy data? (ii) are grazing taxa specialists, or is grazing a means to broaden the dietary niche? and (iii) how is dietary niche breadth attained in taxa at the local level? We demonstrate that while brachydont taxa are specialized as browsers, hypsodont taxa often have broader diets that included more browse consumption than previously anticipated. It has long been accepted that morphology imposes limits on the diet; this synthesis supports prior work that herbivores with 'grazing' adaptions, such as hypsodont teeth, have the ability to consume grass but are also able to eat other foods. Notably, localized dietary breadth of even generalist taxa can be narrow (approx. 30 to 60% of a taxon's overall breadth). This synthesis demonstrates that 'grazing-adapted' taxa are varied in their diets across space and time, and this flexibility may reduce competition among ancient herbivores.

## 1. Introduction

The appearance and expansion of grassland biomes in North America during the Neogene coincided with suites of morphological adaptations and niche diversification among terrestrial mammalian herbivores [1–5]. Most notable is the evolution of horses, from small forms with multiple digits during the Palaeogene, to large taxa with broader muzzles, high-crowned teeth and increased cursoriality [6–10]. This transition, in horses, is cited as one of evolution's classic examples of the ability of animals to adapt to their local environment—with direct selection of features that affect an individual's ability to consume grass.

While morphology can provide insight into an animal's potential dietary behaviour [7,11], it is also evident that certain morphological forms do not necessitate specific diets [12,13]. Once ungulates exhibit higher-crowned teeth, this allows for the inclusion of grass and other abrasive food items into their diet; however, mixed-feeders maintain their ability to consume browse. That being said, browsers are often locked into their diets owing to their physical inability to processes a significant amount of abrasive vegetative material—with fossil tapirs, for example, exhibiting little to no craniodental changes over millions of years [14]. The correlation between craniodental morphology and diet among modern animals has been well established and applied to ecological interpretations of fossil taxa [15,16], but morphology, which is shaped in evolutionary timescales, may not reveal complexities in behavioural or ecological variation within taxa [13,17]. To better understand dietary flexibility and the

degree to which animals consume grass and browse, proxy data from stable carbon isotopes and/or dental microwear are typically employed [18–22]. These data can help infer what an individual animal consumed at a specific place and moment in time, as opposed to its potential diet as inferred from morphological features. However, many critical questions remain unanswered. Specifically: (i) do morphological interpretations of dietary behaviour agree with stable isotope proxy data? (ii) are hypsodont or grazing-adapted taxa specialists, or is hypsodonty a means to broaden the dietary niche? and (iii) how is dietary niche breadth attained at the local level within a taxon?

This paper provides a synthesis of the dietary behaviour and specialization of herbivorous mammals since the expansion of $C_4$ grasslands in North America (the late Neogene). Here, we examined the isotopic record of Perissodactyla, Artiodactyla and Proboscidea occurring at low latitudes (less than 37°) in North America since the late Miocene, approximately 7 Ma (i.e. when and where $C_4$ grasses are favoured, and $\delta^{13}C$ values can more reliably differentiate between the consumption of $C_4$ grass and $C_3$ browse ([18,23,24]; electronic supplementary material). Specifically, we test the following hypotheses of notable relevance to the evolution and palaeobiology of mammalian herbivores through time, focusing on hypsodonty: (i) morphological dietary interpretations largely agree with isotopic proxy data; (ii) the ability to graze does not necessitate a specialized diet of grass; and (iii) the localized isotopic breadth of hypsodont taxa is more variable and flexible than in brachydont taxa.

## 2. Methods

### (a) Materials

Isotopic data include all published stable isotope analyses (SIAs) of carbon from the carbonate portion of tooth enamel hydroxylapatite ($\delta^{13}C$) from herbivorous mammals (i.e. Perissodactyla, Artiodactyla and Proboscidea) since the late Miocene (approx. 7 Ma) that occur in the contiguous United States below 37° latitude. Bulk data (one sample taken parallel to a tooth's growth axis, per individual, typically less than 1 cm in length) and average values from serially sampled teeth (i.e. a series of samples taken perpendicular to a tooth's growth axis) were gathered via a Web of Science search using keywords that included isotope, fossil, teeth and other iterations of these words. Publications where only summary statistics were provided without the raw data are noted in summary tables, but not included in statistical analyses. These data were supplemented with a targeted bulk sampling of under-sampled taxa in poorly sampled regions ($n = 92$; electronic supplementary material, dataset S1). Each occurrence was assigned to a North American land mammal age (NALMA), if known. Each taxon was categorized by hypsodonty index (HI) [15,25] as brachydont (low-crowned), mesodont (moderate-crowned), hypsodont (high-crowned) or highly hypsodont based on published morphological descriptions (gathered from the literature, personal communication with C. Janis, 2020; electronic supplementary material, table S1). All comparisons between taxa occurred at the genus level, owing to concerns over the validity and stability of species-level identifications [26,27]. Presenting results at the genus level also provides taxonomic consistency with many of the original references (electronic supplementary material, dataset S1) and allows for comparisons over deeper time than possible at the species level. This approach is further justified through a hierarchical analysis of variance (taxonomically nested ANOVAS; electronic supplementary material) that

supports the hypothesis that individuals within a genus are congeners with similar diets. Hierarchical analysis of variance was conducted in R using the package 'ape' [28,29].

### (b) Statistical analyses

To assess whether morphological interpretations of dietary behaviour agree with stable isotope proxy data, the isotopic diet was characterized for each taxon. Typical diets of taxa were characterized by calculating the median and interquartile range from available $\delta^{13}C$ values measured from individual specimens. Taxa with median values greater than −2‰ were classified as primarily grazing, values less than −9‰ were classified as primarily browsing, and intermediate values were classified as mixed-feeding [30]. Taxa were secondarily classified by the breadth of the interquartile range. Taxa with median diets where the third quartile was less than −9‰ were classified as 'browsers', while 'browsing/mixed-feeders' had median values less than −9‰, but a third quartile that exceeded this threshold. Taxa were identified as 'grazers' where the third quartile was greater than −2‰ and 'mixed-feeders' were classified for those taxa where the first and third quartiles were between −9‰ and −2‰ or, as was the case with *Camelops*, the interquartile range fully spanned −9‰ to −2‰.

In addition to classifying the overall median diet and breadth of taxa, dietary breadth and specialization were analysed at the local, or site, level within a NALMA. Local breadth is naturally expected to be a fraction of a taxon's overall breadth. To assess whether average local breadth is indeed narrower, or more specialized, than would be expected from a random sampling of the taxon pool, the standardized mean effect size (Cohen's $d$) between a taxon's observed mean site range and the expected mean range was calculated from a set of randomizations (electronic supplementary material).

To assess how generalist taxa acquire their overall breadth and quantify how representative localized samples are of a taxon's overall diet, the local $\delta^{13}C$ range at a site was calculated as a fraction of the overall taxon $\delta^{13}C$ range. The average site fraction was compared by taxon to assess whether generalists with broad diets tended to also be generalists at the local level. If a site was composed of assemblages of differing NALMAs, fractions were calculated for each, separately. Only assemblages with five or more individuals of that taxon were included in this analysis, and a mean fraction was calculated if it was found within greater than or equal to three assemblages.

The localized dietary breadth of adaptable taxa may be impacted by biotic interactions, such as competition and resource availability. If hypsodonty affords species greater flexibility in their diets, then hypsodont taxa are predicted to obtain greater breadth at the local level through the consumption of alternative (non-grass) resources, while brachydont taxa are not. Local dietary breadth (the range of $\delta^{13}C$ values) of each taxon was fit as a linear function of its site minimum and maximum. If a site was composed of assemblages of differing NALMAs, each assemblage was treated separately. Only assemblages with five or more individuals of that taxon were included in this analysis, and a regression was fit if it was found at greater than equal to three assemblages. Relationships were assessed by taxa, by HI categories, and by isotopically determined dietary categories (browser, browsing/mixed-feeder, mixed-feeder, grazer). Statistical analyses and data visualizations were conducted in 'R' using the packages 'plyr' and 'ggplot2' [28,31,32].

## 3. Results

### (a) Dietary consensus and specialization

All compiled data of stable carbon isotope values from fossil mammals ($n = 1312$; 1161 when excluding California;

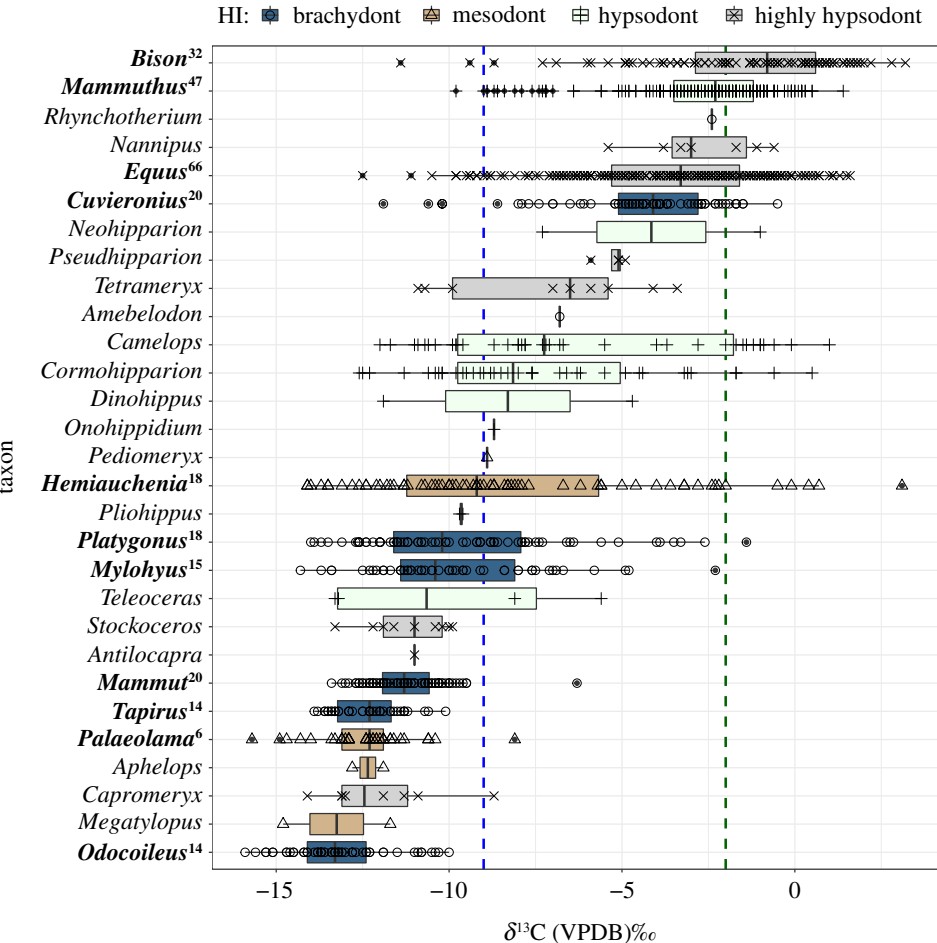

**Figure 1.** Comparison of diets across 28 fossil mammal taxa ($n = 1161$) showing the medians and interquartile ranges (boxes) of $\delta^{13}C$ values from fossil tooth enamel. These individuals are from localities below 37° latitude and outside of the state of California. Taxa are arranged from bottom to top in order of increasing median $\delta^{13}C$ values. Box-colour indicates relative hypsodonty: brachydont indicated by dark blue bars with open circles, mesodont indicated by tan bars with open triangles, hypsodont indicated by light green bars with plus symbols and highly hypsodont indicated in grey with x's. The 11 taxa that are best represented in the dataset (at least three distinct assemblages with at least five individuals) are labelled in bold, with their total number of sites indicated as a superscript. Values less than -9‰ (dashed blue line) are diets of primarily $C_3$ resources (browse, at low latitudes), while values greater than -2‰ (dashed green line) are diets of primarily $C_4$ resources (grass, at low latitudes). (Online version in colour.)

electronic supplementary material, dataset S1, [33]) are summarized in figures 1 and 2 and table 1 (electronic supplementary material, figures S6 and S7, and tables S2 and S3). Out of the 30 taxa with published $\delta^{13}C$ values in the literature (electronic supplementary material, table S1), 29 occur outside of California (figures 1 and 2; electronic supplementary material table S3), and 11 are well represented (found at $n \geq 3$ sites with $n \geq 5$ individuals; $n = 658$ individuals samples, across the best-sampled localities, table 1). Of these, HI indicates that *Bison* and *Equus* are highly hypsodont; *Mammuthus* is hypsodont; *Hemiauchenia* and *Palaeolama* are mesodont; and *Mylohyus*, *Platygonus*, *Tapirus*, *Cuvieronius*, *Mammut*, and *Odocoileus* are brachydont (electronic supplementary material, table S1). There is an agreement between SIA data and most brachydont taxa (figure 1 and table 1), and the narrow breadth of their typical diets (interquartile range) indicates that these taxa are specialized on $C_3$ vegetation (i.e. browse, at sites less than 37° latitude). *Curvieronius* is an exception, as it has a median $\delta^{13}C$ value greater than −9‰ but less than −2‰. *Mylohyus* and *Platygonus* have median diets that are less than −9‰, but they are distinguished from other brachydont taxa in the broad interquartile breadth of their $\delta^{13}C$ values, indicating they are browsers/mixed-feeders. Among hypsodont

taxa, *Bison* is the only one with a median $\delta^{13}C$ value greater than −2‰, indicating a primarily grazing diet; however, the breadth of its interquartile range indicates that *Bison* often consume $C_3$ vegetation (i.e. potentially browse; figure 1 and table 1). *Mammuthus* and *Equus* (hypsodont and highly hypsodont, respectively) have median values ≤−2‰, but the breadth of their diets does indicate significant consumption of $C_4$ resources (figure 1 and table 1). *Palaeolama* and *Hemiauchenia* are both mesodont and are browsing and browsing/mixed-feeding, respectively. Although not well-sampled outside of California (electronic supplementary material, table S2), *Camelops*, *Cormohipparion* (both hypsodont), *Tetramaryx* and *Nannipus* (both highly hypsodont) are, isotopically, mixed-feeders, and *Stockoceros*, which is primarily sampled within California, is a highly hypsodont browser (figure 1; electronic supplementary material, figures S6 and S7). Among well-sampled taxa ($n = 11$) there are significant, positive, linear relationships between both a taxon's median $\delta^{13}C$ value ($R^2 = 0.3046$, $p = 0.0455$) and mean $\delta^{13}C$ value ($R^2 = 0.3916$, $p = 0.0233$), and their total $\delta^{13}C$ range. There is a significant positive relationship between a taxon's dietary range and their third quartile value ($R^2 = 0.5011$, $p = 0.0089$) and a highly significant relationship with a taxon's maximum $\delta^{13}C$

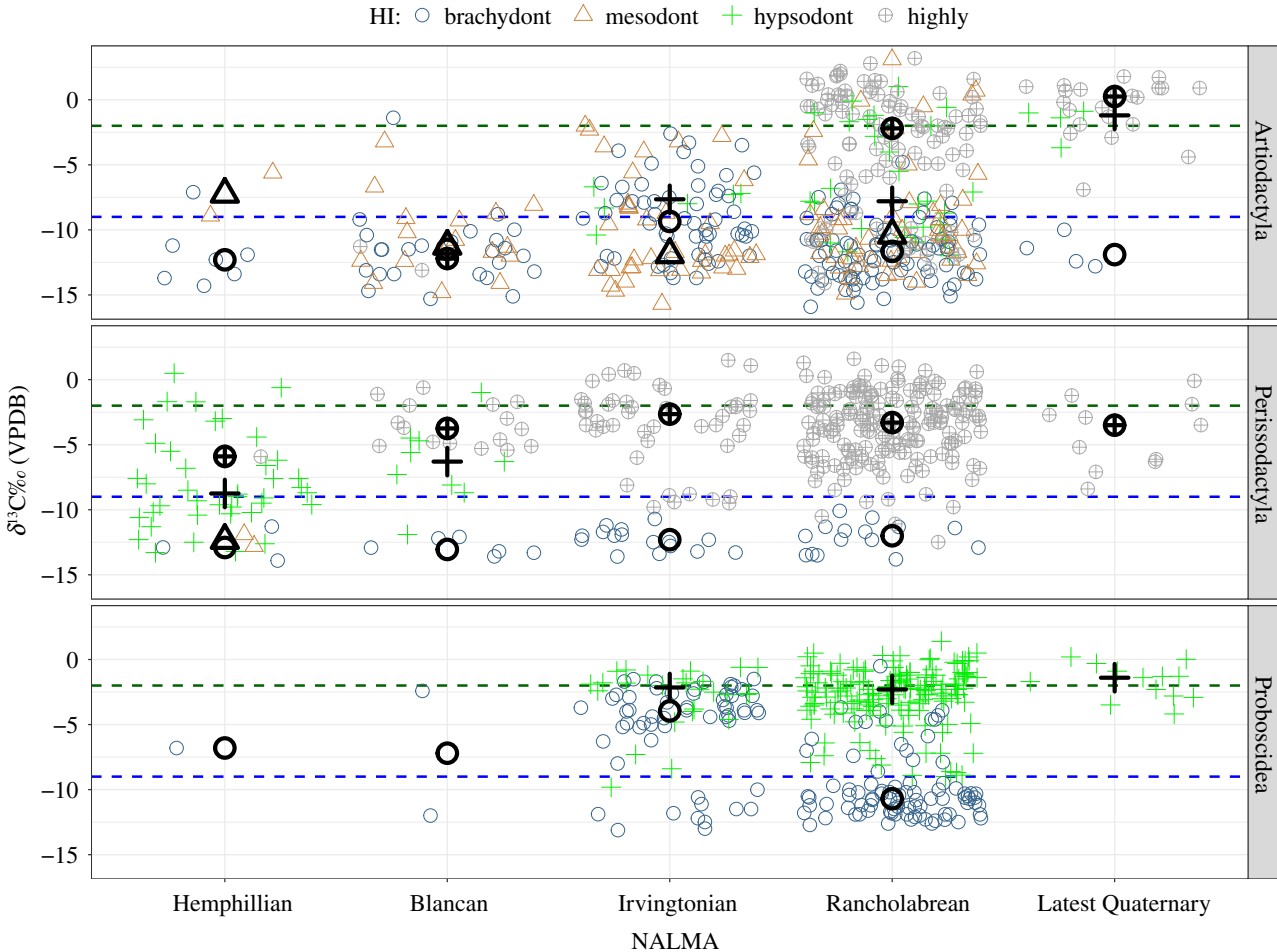

**Figure 2.** Jitter plot of dietary variation among taxa with brachydont, mesodont, hypsodont and highly hypsodont dentitions, within orders, over time ($n = 1138$). All data included are from localities outside of California, below 37° latitude. Time bins are North American land mammal ages (NALMA), with the oldest record specimens occurring in the Hemphillian, approximately 7 Ma. Absolute positions along the *x*-axis within time bins are random: points are horizontally jittered within time bins for visual clarity. Median values for each dietary category are indicated with bold symbols for each NALMA. (Online version in colour.)

**Table 1.** Comparison between hypsodonty index (HI) and diets inferred through SIA of $\delta^{13}C$ values from fossil herbivore tooth enamel. (HI categories are brachydont (B), mesodont (M), hypsodont (H) and highly hypsodont (HH). SIA diet categories are grazer (G), mixed-feeder (MF), browser/mixed-feeder (BMF) and browser (B). The total number of specimens and sites ('*n*(*n* sites)') are provided in addition to the sample sizes from sites that have at least five individuals present (*n* of best). Summary statistics for each taxon are provided, including the median, mean with standard deviation, total range and interquartile range (IQR). SIA diet is primarily determined by the median and is modified by the breadth indicated from the IQR. Taxa are indicated in bold if the primary, median, diet differs from what is expected from the HI.)

| taxon | HI | SIA | *n*(*n* sites) | *n* of best | med. (‰) | $\bar{x} \pm$ s.d. (‰) | range (‰) | IQR (‰) |
|---|---|---|---|---|---|---|---|---|
| ***Cuvieronius*** | **B** | **MF** | 69(20) | 45(4) | −4.1 | −4.5 ± 2.3 | −11.9 to −0.5 | −5.1 to −2.8 |
| *Mammut* | B | B | 84(20) | 62(7) | −11.3 | −11.2 ± 1.1 | −13.4 to −6.3 | −11.9 to −10.6 |
| ***Mylohyus*** | **B** | **BMF** | 50(15) | 33(3) | −10.4 | −9.8 ± 2.5 | −14.3 to −2.3 | −11.4 to −8.1 |
| *Odocoileus* | B | B | 57(14) | 43(6) | −13.3 | −13.2 ± 1.4 | −15.9 to −10 | −14.1 to −12.4 |
| ***Platygonus*** | **B** | **BMF** | 74(18) | 42(4) | −10.2 | −9.6 ± 2.8 | −14 to −1.4 | −11.6 to −7.9 |
| *Tapirus* | B | B | 40(14) | 25(4) | −12.3 | −12.4 ± 1 | −13.9 to −10.1 | −13.2 to −11.7 |
| *Hemiauchenia* | M | BMF | 72(18) | 50(5) | −9.2 | −8.3 ± 4 | −14.1 to 3.1 | −11.2 to −5.7 |
| ***Palaeolama*** | **M** | **B** | 37(6) | 31(3) | −12.3 | −12.4 ± 1.4 | −15.7 to −8.1 | −13.1 to −11.9 |
| *Mammuthus* | H | G | 196(47) | 135(13) | −2.3 | −2.7 ± 2.1 | −9.8 to 1.4 | −3.5 to −1.2 |
| *Bison* | HH | G | 98(32) | 54(7) | −0.8 | −1.4 ± 2.7 | −11.4 to 3.2 | −2.9 to 0.6 |
| *Equus* | HH | G | 237(66): | 138(13) | −3.3 | −3.6 ± 2.7 | −12.5 to 1.6 | −5.3 to −1.6 |

value ($R^2 = 0.8794$, $p < 0.0001$). Collectively, these data indicate a positive relationship between grass consumption and overall dietary breadth.

Across time, diets are constrained among brachydont Artiodactyls and Perissodactyls, while more flexible diets are associated with hypsodonty (figure 2). Although there is

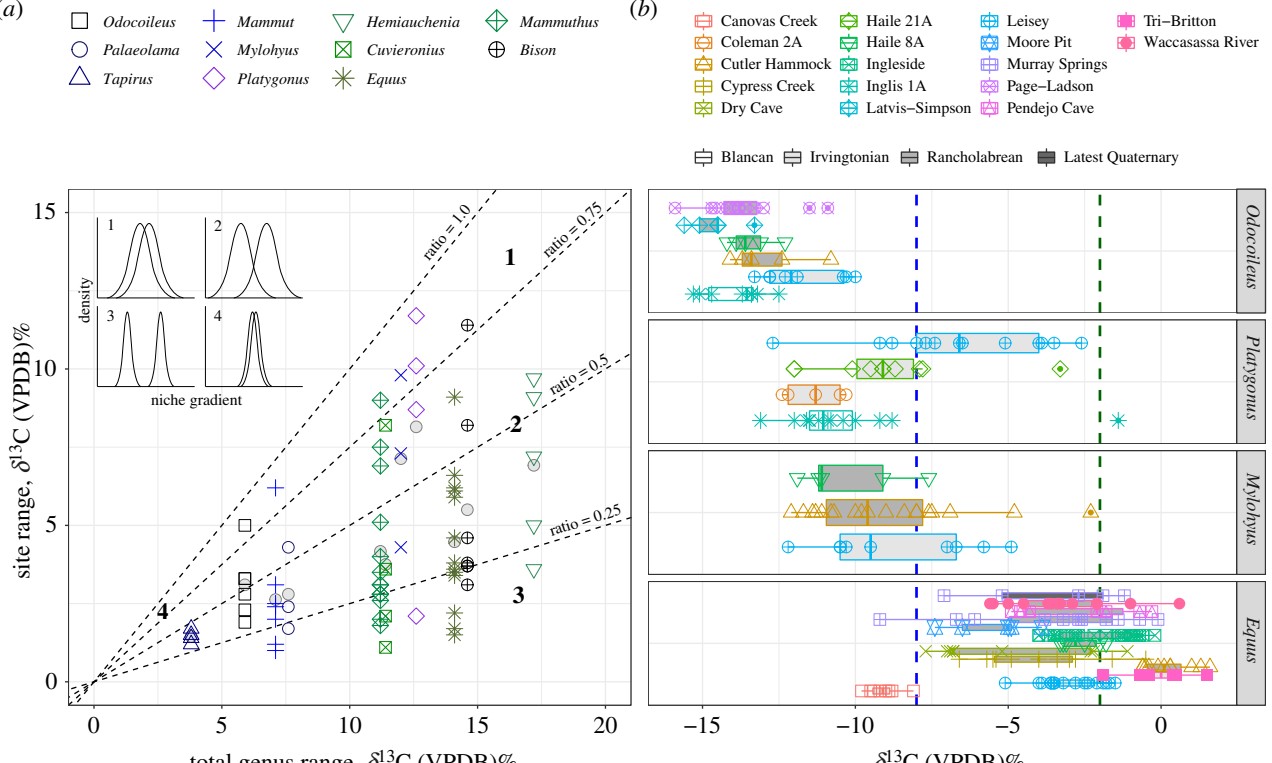

**Figure 3.** Localized breadth at localities with at least five individuals present. (*a*) The local breadth of a taxon site (the mean indicated by a grey circle) is typically a small fraction (less than 0.5) of its overall breadth. The insert (adapted from Bolnick *et al.* 2003 [34]) shows a conceptual depiction of how individual populations partition the overall niche breadth of a taxon. Most taxa in this study have ratios between 0.32 and 0.60, corresponding to conceptual depictions 2 or 3, where populations are different, but can overlap somewhat in their breadth. (*b*) Site specific values presented as boxplots for taxa illustrate the relatively narrower breadth at the local level and the distinctions between 'populations'. Site names are indicated by marker shapes. The NALMA of a site is indicated by the shading of the bar. In this example, the average fraction of the total dietary breadth across sites for *Equus* (a highy hypsodont grazer; $n = 138$ across 13 localities) is less than *Mylohyus* (a brachydont browsing mixed-feeder; $n = 33$ across three localities), *Platygonus* ($n = 42$ across four localities) and *Odocoileus* (a brachydont browser; $n = 43$ across six localities). (Online version in colour.)

variation over space (electronic supplementary material, figure S8), the overall pattern is similar. Median grazing values greater than −2‰ are achieved by highly hypsodont Artiodactyls by the Rancholabrean and hypsodont Proboscideans in the Latest Quaternary. This is true even if a geographical region is considered (electronic supplementary material, figure S8). With the exception of some southwestern Irvingtonian occurrences (electronic supplementary material, figure S8), median diets of Perissodactyls remain less than −2‰ since throughout the Pleistocene and Latest Quaternary. Mesodont taxa have median diets reflective of browsing (< −9‰) since the Blancan. The median diets of brachydont Artiodactyls and Perissodactyls are reflective of browsing, while this morphology is associated with mixed-feeding among Proboscideans until the Latest Quaternary, when *Cuvieronius* goes extinct.

Specialization at the locality level varies across taxa (figure 3) and average site breadth is provided in table 2. *Equus*, *Hemiauchenia* and *Tapirus* have average site breadths that are more constrained than expected from random (table 2; electronic supplementary material, table S4), as indicated by standardized mean effect sizes that are large ($d >$ 0.8). The average fraction of the total taxon breadth ranges from $0.32 \pm 0.16$ s.d., for the least locally representative taxon (*Equus*) relative to its overall breadth, to the most representative (*Platygonus*) at $0.65 \pm 0.31$ (figure 3*a* and table 2). Among the best-represented taxa, there is no significant linear relationship between a median $\delta^{13}$C diet and the average fraction of the overall breadth that is represented locally at a site ($p = 0.1574$). Site fractions for *Odocoileus* (a brachydont browser)

are significantly higher than *Equus* ($p = 0.0295$; a highly hypsodont grazer; figure 3*b*). The fraction of overall breadth that is locally represented by *Platygonus* and *Mylohyus* (brachydont browsers/mixed-feeders) are also significantly higher than *Equus* ($p = 0.0485$ and $p = 0.0426$, respectively; figure 3*b*).

## (b) Dietary breadth

The relationships between localized dietary breadth and minimum and maximum $\delta^{13}$C values are summarized in figure 4. Among *Cuvieronius* ($p = 0.0157$), *Mammuthus* ($p < 0.0001$) and *Bison* ($p = 0.0029$), there is a significant relationship between greater dietary breadth and lower (depleted) $\delta^{13}$C values (greater consumption of $C_3$ resources; figure 4*a* and table 2). The localized breadth of *Mammut* ($p = 0.0056$), *Platygonus* ($p = 0.0052$) and *Mylohyus* ($p = 0.0265$) are significantly broader with higher (enriched) $\delta^{13}$C values (figure 4*b* and table 2). Collectively, hypsodont taxa increase their breadth with greater consumption of $C_3$ resources ($p < 0.0001$; figure 4*a*; electronic supplementary material, table S5) and brachydont taxa increase their breadth with greater consumption of resources with enriched $\delta^{13}$C values ($p < 0.0001$; figure 4*b*). If we consider the categorization of taxa based on their median isotopic values and breadth, browsers significantly increase their local dietary breadth with greater consumption of resources with enriched $\delta^{13}$C values ($p = 0.0026$; figure 4*d*; electronic supplementary material, table S5). There is no significant relationship between local dietary breadth and the maximum or minimum local value for browsing/mixed-feeders. By contrast, mixed-feeders

**Table 2.** Locality parameters pertaining to the breadth of each taxon. (The average assemblage range and the standard deviation were calculated where there were at least five individuals present (*n* of best) with the number of sites given in parentheses. The standard mean effect size (Cohen's *d*) between the mean assemblage range and the mean expected from randomizations was calculated, and large effect sizes ($d \geq 0.80$) indicated in bold. The local breadth ($\delta^{13}$C range) of a taxon divided by its total range (table 1), presented as the fraction ratio across all sites ± s.d. Significance values between a taxon's local breadth and its local maximum and minimum are also given. '—'indicates that fewer than three sites were available to calculate a relationship, and values in bold are significant linear relationships.)

| taxon | n of best | range ± s.d. (‰) | d | fraction ± s.d. (‰) | p-values, relationship local breadth | |
|---|---|---|---|---|---|---|
| | | | | | taxon min | taxon max |
| *Bison* | 54(7) | 5.5 ± 3.1 | 0.33 | 0.38 ± 0.21 | **0.0029** | 0.8484 |
| *Mammuthus* | 135(13) | 4.2 ± 2.3 | 0.27 | 0.37 ± 0.20 | **0.0001** | 0.2168 |
| *Equus* | 138(13) | 4.5 ± 2.2 | **0.85** | 0.32 ± 0.16 | 0.0919 | 0.2414 |
| *Cuvieronius* | 45(4) | 3.8 ± 3.1 | 0.28 | 0.33 ± 0.28 | **0.0157** | 0.2874 |
| *Hemiauchenia* | 50(5) | 6.9 ± 2.6 | **0.91** | 0.40 ± 0.15 | 0.2447 | 0.7213 |
| *Mammut* | 62(7) | 2.6 ± 1.7 | 0.08 | 0.37 ± 0.24 | 0.4217 | **0.0056** |
| *Mylohyus* | 33(3) | 7.1 ± 2.8 | 0.66 | 0.59 ± 0.23 | 0.5122 | **0.0265** |
| *Odocoileus* | 43(6) | 3.1 ± 1.1 | 0.03 | 0.53 ± 0.18 | 0.5814 | 0.1799 |
| *Palaeolama* | 31(3) | 2.8 ± 1.3 | 0.75 | 0.37 ± 0.18 | 0.5202 | 0.2128 |
| *Platygonus* | 42(4) | 8.2 ± 4.2 | 0.14 | 0.65 ± 0.31 | 0.5158 | **0.0052** |
| *Tapirus* | 25(4) | 1.5 ± 0.2 | **0.83** | 0.38 ± 0.05 | 0.7874 | 0.9605 |
| *Tetrameryx* | 7(1) | 7.3 ± — | — | — | — | — |
| *Camelops* | 15(2) | 8.4 ± 6.6 | 0.00 | 0.64 ± 0.51 | — | — |
| *Cormohipparion* | 33(1) | 13.1 ± — | — | — | — | — |
| *Stockoceros* | 6(1) | 3.4 ± — | — | — | — | — |
| *Nannipus* | 6(1) | 4.8 ± — | — | — | — | — |

($p = 0.0072$) and grazers ($p < 0.0001$) increase their local dietary breadth by individuals in the population consuming more $C_3$ resources (figure 4*c*; electronic supplementary material, table S5.

## 4. Discussion

### (a) Dietary consensus and specialization

Palaeoecological interpretations that are based on faunal composition rely on a precise understanding of the dietary and habitat preferences of fossil taxa. The degree of hypsodonty was compared to $\delta^{13}$C values of enamel, an isotopic proxy for diet. Many brachydont taxa, as expected, have diets categorized by $\delta^{13}$C values as browsers (figures 1 and 2). However, SIA refines the diets of mesodont–hypsodont taxa and provides greater insight regarding their dietary breadth (figure 1), and variability over time (figure 2) and space (figures 3 and 4). Notably, $\delta^{13}$C values indicate some degree of mixed-feeding behaviour, or $C_3$ consumption, across mesodont to highly hypsodont taxa (figures 1 and 2 and table 1). Hypsodonty and the ability to consume $C_4$ resources are associated with greater isotopic breadth (figure 1) and the ability to consume $C_3$ resources over time (figure 4). Although dietary breadth at the local level represents a fraction of the overall dietary breadth of a taxon, that fraction is most representative of the overall diet for taxa such as *Odocoileus* (a brachydont browser) and *Mylohyus* (a brachydont browser/mixed-feeder) than *Equus* (a highly hypsodont grazer). Although the fraction of their overall

dietary breadth represented at the local level is not significantly different from other taxa, *Tapirus* and *Hemiauchenia* have average site ranges that are specialized (narrower than expected from random; table 2).

These data suggest that the brachydont tooth morphology 'locks' most taxa into specialized browsing behaviour, while hypsodonty and the ability to graze allows many taxa to be more generalized and flexible—increasing dietary plasticity. Notable exceptions are the brachydont peccaries, *Mylohyus* and *Platygonus*, which take advantage of resources with enriched $\delta^{13}$C values, and the largest brachydont taxon in this study, *Cuvieronius*, which is a mixed-feeder. The common representation of Pleistocene *Mammuthus*, *Equus* and *Bison* as 'grazing specialists' in the literature [2,35–37] stands in contrast with their broad breadth of $\delta^{13}$C values presented here.

While it is possible that a portion of the breadth observed among grazers is owing to the consumption of $C_3$ grass, we sought to mitigate the confounding effects of $C_3$ grasses by restricting our analyses to occurrences found in regions where $C_4$ grasses are the dominate grass species (<37° latitude) [18,23,24,35,38] (electronic supplementary material). Despite this constraint, many of the taxa in our study are large and probably had large home ranges. Additionally, some of the late Hemphillian records may reflect a greater abundance of $C_3$ grasses that were present before the development of $C_4$-dominated grassland ecosystems [17]. Slightly depleted $\delta^{13}$C values could result from an animal moving between $C_4$-grass- and $C_3$-grass-dominated environments. However, to

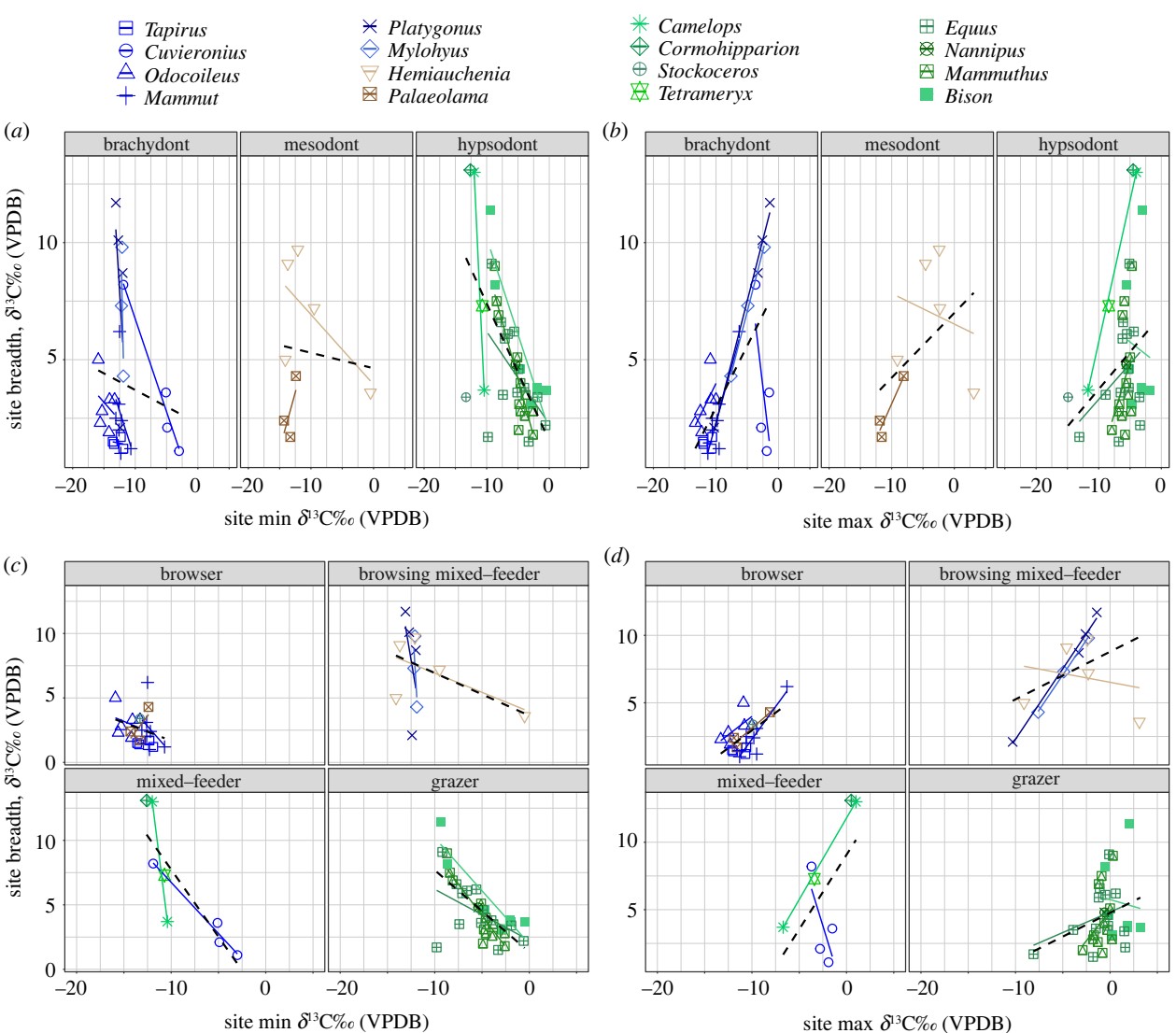

**Figure 4.** A taxon's local dietary breadth as a function of consuming varied resources. Taxa are categorized by hypsodonty (*a,b*) and their isotopically determined diets (*c,d*). (*a*) The local dietary breadth of hypsodont taxa increases when individuals consume more $C_3$ resources ($n = 39$, $p = 3.80 \times 10^{-0.6}$), while (*b*) the local dietary breadth of brachydont taxa increases when individuals consume more $C_4$ resources ($n = 28$, $p = 7.12 \times 10^{-05}$). Among different dietary guilds, (*c*) browsers ($n = 21$, $p = 0.0026$) exhibit greater dietary breadth when more $C_4$ is consumed, and (*d*) mixed-feeders ($n = 8$) and grazers ($n = 34$) exhibit greater dietary breadth with greater consumption of $C_3$ resources ($p = 0.0072$, $p = 4.88 \times 10^{-06}$, respectively). When both categories of mixed-feeders are combined, local dietary breadth is significantly associated with greater consumption of $C_3$ resources ($n = 20$, $p = 0.0036$). (Online version in colour.)

obtain some of the lower first quartile $\delta^{13}C$ values observed for *Equus* (−5.4‰), large quantities of $C_3$ grasses would need to be consumed via migrations to $C_3$-grass-dominated environments (electronic supplementary material). Under such a scenario, isotopic values would show large variation over seasons and could be revealed by serially sampling teeth along their growth axis. However, a supplementary analysis on a subset of serially sampled individuals from low-latitude North America since 5 Ma does not support this alternative (electronic supplementary material, figure S9). The abundance of $C_3$-grass is controlled by precipitation and temperature; therefore, a low-latitude longitudinal environmental gradient would tend to favour $C_4$ over $C_3$ grasses in eastern versus western sites [38,39]. As a result, grass consumed from western sites may result in enamel with lower $\delta^{13}C$ values. However, the data in our study tend to be temporally and spatially biased to Quaternary sites in the southeastern United States. Within this region, and during this time, grazers would have to engage in suboptimal, selective grazing of relatively rare $C_3$ grass, to the exclusion of more readily abundant $C_4$ resources.

Furthermore, when analysed in isolation, eastern sites result in similar dietary characterizations across taxa (electronic supplementary material, figures S6 and S7, tables S2 and S3) and over time (electronic supplementary material, figure S8). Thus, we interpret the observed breadth of $\delta^{13}C$ values of grazing-adapted taxa as a true, mixed-feeding signal.

SIA helps to confirm dietary flexibility observed among herbivorous taxa, as suggested via other proxies. The $\delta^{13}C$ values of *Platygonus* and *Mylohyus* verify their primary diet as that of a browser, as would be expected from their dental and cranial morphology [40,41], yet their broad diets include mixed-feeding, consistent with dental microwear analyses [42,43]. Further, evidence from dental microwear analysis indicates that even grazing-adapted taxa, such as *Mammuthus* [44] and *Bison* [19,45], exhibit varied diets that include forbs and shrubs. While dental microwear texture analysis (DMTA) is often cited as recording the most recent meals an animal consumes [46], sampling of multiple individuals that died at disparate periods of time and across disparate age classes demonstrates the overwhelming

consumption of browse among grazing-adapted taxa in fossil assemblages. The high median $\delta^{13}C$ values (e.g. −3.3‰ for *Equus*) and broad interquartile ranges (e.g. 3.6‰ for *Bison*) we report here would suggest that the variation in dietary textures observed through DMTA is not merely capturing the consumption of fallback resources or the 'last supper' of a taxon before its death, but prolonged mixed-feeding behaviour at the time teeth was mineralizing (as well as shortly before death).

Although many taxa exhibit a wide range of $\delta^{13}C$ values, there is evidence of site specialization among some taxa (figure 3 and table 2). These findings are consistent with other studies addressing intraspecific variation over space [13,17]. Just as specialization among individuals within a population can have important ecological implications [34], specialization across populations within a taxon may reveal important insights into resource use and competition within communities over broader scales [17]. Among three of the 11 well-represented taxa in our study, the average local breadth is lower than expected from random (table 2, $d > 0.8$). Of these, *Hemiauchenia* has the broadest overall dietary breadths (interquartile range = 5.5‰, table 1) of the best-represented taxa in our study, but the average fraction of its dietary breadth is low, $0.40 \pm 0.15$ (table 2). *Hemiauchenia* therefore represents a taxon that is overall a generalist, but rather than adopting a jack-of-all-trades strategy at all sites, it is often fairly specialized at a given locality.

Across most taxa, local dietary breadth accounts for less than half of a taxon's overall dietary breadth (average local fraction <0.5; figure 3a). There is no significant relationship between median diet and the average fraction of dietary breadth represented at the local level. However, apparent differences in the localized ratios of some grazers (i.e. *Equus*) compared to some browsers and browser/mixed-feeders (i.e. *Odocoileus*, *Platygonus* and *Mylohyus*, figure 3b), hint that dietary partitioning at the local level may be influenced by dietary strategy. Somewhat surprisingly, localized specialization is also evident among comparatively specialized browsers (e.g. *Tapirus*; $d > 0.8$, table 2). The two extinct peccaries (*Mylohyus* and *Platygonus*) have broad diets and appear to be the most representative of their greater breadth at the local level. This contrasts with modern observations of *Tayassu pecari*, which are narrowly dependent (−28.7‰ to −26.9‰, $\delta^{13}C$) on closed-environment $C_3$ resources across Brazilian biomes [47]—suggesting that the study of extinct taxa can alter our understanding of dietary plasticity among herbivorous mammals.

## (b) Grazing morphology as a means to broaden the niche

Given evolutionary adaptations that allow for grass consumption, it was predicted that greater localized dietary breadth among hypsodont taxa, mixed-feeders and grazers would be associated with more consumption of $C_4$ resources. We found that, as a guild, hypsodont taxa significantly increase their localized dietary breadth when they consume more $C_3$ resources (figure 4a and table 2). Similarly, among grazers and mixed-feeders, greater breadth was associated with lower $\delta^{13}C$ values, or greater $C_3$ consumption (figure 4c and table 2). This suggests that morphological adaptations associated with grazing (e.g. hypsodont and loxodont tooth morphologies or high post-canine tooth volume) do not

exclude opportunistic expansions of the dietary niche to consume forbs and shrubs [12,13,35,48,49], as opposed to mandating grazing. The observed amount of browse consumption among highly hypsodont taxa raises the question of what importance grit may play with extinct grazers, especially *Equus*. Given that inadequate consumption of abrasives in the diets of modern horses can result in pathological uneven tooth wear [50], future directions of the dietary niches of extinct *Equus* might evaluate minimum required grit.

Brachydont and browsing taxa exhibited greater dietary breadth in association with higher $\delta^{13}C$ values than was predicted (figure 4b,d). Although it is clear that these animals are not consuming large quantities of grass (figure 1), these relationships do suggest that they are partitioning $C_3$ resources, which are more isotopically variable than $C_4$ resources, to exploit more $C_3$ vegetation with enriched $\delta^{13}C$ values, possibly in more open habitats [18]. The ability for a taxon to broaden its niche, or to engage in variable dietary behaviour, may allow a buffer against extinction. As Janis *et al.* [1] point out, there is a large decline of browsers with the expansion of grassland biomes in North America, but not a one-for-one replacement of grazers over browsers since the late Miocene. This shift in dietary strategy probably permitted grazing-adapted or mixed-feeding taxa to now eat a broader range of resources depending on climatic conditions, the presence of competitors and other biotic and/or abiotic factors. Given this apparent flexibility, we suggest a re-phrasing of dietary categorizations from grazing to 'grazing-adapted' and mixed-feeding to 'mixed-feeding-adapted' moving forward.

## (c) Lost ecological functions, biodiversity loss and underlying interactions within communities

Any interpretation of the past using modern analogues relies on the principle of uniformitarianism: that the processes occurring in the present day have always occurred. When inferring diets of long-extinct species, we often rely on morphological comparisons of features that have evolved over millennia and their correlations with observed dietary behaviour [7,11]. We also assume that what an animal does today, it probably did so in the past and will also do so in the future (i.e. niche conservatism [51]). We know, however, from separate lines of evidence, that ecological interactions and realized niches are not uniform through space or time. Notably, the study of non-analogous mammalian communities has revealed that taxa respond individualistically in response to environmental change [52–55], and populations within taxa are also known to behave individualistically and respond locally to the environment [56–59].

Mismatch between diets inferred via $\delta^{13}C$ values and from extant morphological comparisons may also reveal fundamental differences between ecological communities of the prehistoric past and the present. Our results suggest that community interactions might be driving niche breadth and partitioning within sites, which would be expected among diverse communities with many types of herbivores. However, modern systems are significantly less diverse than those of the past, and many niches that were filled as recently as *ca* 13 000 yr B.P. (such as grazing-adapted megafauna) are now absent, or nearly absent, from most ecosystems, including in North America. The extinction of megafauna has been linked to the complete reorganization of interactions between organisms, such that species with strong community

associations in the past are now significantly disassociated in modern communities [60–63]. Therefore, our observations of modern communities, and perceptions of specialization across taxa in these environments, may not serve as suitable analogues for past communities.

It has long been accepted that morphology provides a first approximation of diet. While morphology may limit what an animal can and cannot eat (i.e. brachydont taxa do not extensively graze, and carnivores with gracile jaws do not regularly crush bone), morphology does not necessarily prescribe a specialized diet [12,13,35,49]. The use of proxy data such as stable isotopes and dental microwear arose because of the need to understand what individual animals actually ate at a given location and point in time [18,21,46]. Collectively, this synthesis of stable carbon isotope data helps provide an informed estimate of the dietary behaviour of 30 taxa, and detailed analyses of those best represented in the literature permit an assessment of dietary breadth, specialization and variation across localities. This synthesis collectively supports the idea [25,48] that herbivores with 'grazing' adaptions have exactly that—the ability to consume grass but are also able to eat browse and/or other foods depending on local factors. A re-framing of terms such as grazers and mixed-feeders to 'grazing-adapted' and 'mixed-feeding adapted', respectively, may help clarify the palaeobiology and evolutionary history of ancient herbivorous mammals in North America.

Data accessibility. Isotope data files are available from the Dryad Digital Repository: https://doi.org/10.5061/dryad.r4xgxd2bj [33].

Authors' contributions. M.I.P. participated in the design of the study, carried out specimen sampling, conducted the data analyses, and drafted and revised the manuscript; L.R.G.D. participated in the design of the study, carried out specimen sampling and helped critically revise the manuscript.

Competing interests. We declare we have no competing interests.

Funding. This work was supported by the National Science Foundation (EAR1725154) and Vanderbilt University.

Acknowledgements. We thank A. Millhouse of the Smithsonian National Museum of Natural History, G. Morgan of the New Mexico Museum of Natural History and Science, and R. Hulbert of the Florida Museum of Natural History for their expertize, collections access and assistance with sampling specimens. We thank J. Curtis of the Department of Geosciences at the University of Florida for timely analyses of stable isotopes and we thank C. Janis for discussions regarding hypsodonty characterizations. The authors are appreciative of the thorough feedback received from two anonymous reviewers and associate editor J. Hutchinson. M.I.P. was generously hosted by the departments of Earth and Environmental Sciences and Biological sciences at Vanderbilt University while conducting this work.

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
