## [Peer Review File · Proceedings of the Royal Society B: Biological Sciences]

Review History

RSPB-2020-0945.R0 (Original submission)

Review form: Reviewer 1

Recommendation

Major revision is needed (please make suggestions in comments)

Scientific importance: Is the manuscript an original and important contribution to its field?

Excellent

General interest: Is the paper of sufficient general interest?

Excellent

Quality of the paper: Is the overall quality of the paper suitable?

Good

Is the length of the paper justified?

Yes

Should the paper be seen by a specialist statistical reviewer?

No

Do you have any concerns about statistical analyses in this paper? If so, please specify them explicitly in your report.

No

It is a condition of publication that authors make their supporting data, code and materials available - either as supplementary material or hosted in an external repository. Please rate, if applicable, the supporting data on the following criteria.

Is it accessible?

Yes

Is it clear?

Yes

Is it adequate?

Yes

Do you have any ethical concerns with this paper?

No

Comments to the Author

Pardi and DeSantis have presented a very interesting study that adds significantly to our understanding of the feeding ecology of Neogene North American herbivores. I think the study is of appropriate relevance and broad interest to be considered for publication in the Proceedings of the Royal Society B. However, I have several comments and concerns that I would like the authors to consider. My concerns and comments are detailed below.

Morphologically-inferred diets

There are inconsistencies in the morphological traits that the authors cite as evidence to morphologically infer the diet of the taxa in their study. Based on the references provided in Table S1, some of the taxa are assigned to a dietary category based on evidence from dental microwear (Capromeryx, Cormohipparion, Stockoceros, and Tetrameryx), degree of hypsodonty (13 out of the 30 taxa studied), premaxillary shape (all camelids), changes in enamel arrangement and/or structure (Cuvieronius and Mammuthus), and multivariate assessment of craniodental features (Pediomeryx). In my opinion, the application of dental microwear and degree of hypsodonty to morphologically infer the diet of the taxa in the study are the most problematic and my concerns are expressed below.

The authors assigned the extinct antilocaprids (Capromeryx, Stockoceros, and Tetrameryx) to a dietary category based on the study by Semprebon and Rivals (2007). Semprebon and Rivals (2007) made dietary inferences for those taxa based on their analysis of dental microwear. As Pardi and DeSantis point out in the Introduction of their manuscript, dental microwear and stable isotopes provide insights into “what an individual animal consumed at a specific place and moment in time, as opposed to its potential diet as inferred from morphological features...” (Pardi and DeSantis page 4). Therefore, the authors should reassess the dietary assignment of the extinct antilocaprids using proper morphological traits.

For the equids Cormohipparion, Hipparion, and Onohippidium the authors cite MacFadden (2005) as the source of their morphologically assigned diet. MacFadden (2005) presents a figure that summarizes the phylogeny, geographic distribution, diet, and body sizes of the family Equidae at the generic level. In this figure, Cormohipparion is identified as a mixed-feeder, Hipparion as a grazer, and Onohippidium as a mixed-feeder. MacFadden (2005) does not explicitly state what kind of data was used to infer the diet of the different genera shown in the figure. However, in the text he mentions that some hypsodont equids were grazers and others mixed-feeders, citing the study by MacFadden et al. (1999). The study by MacFadden et al. (1999) inferred the diets of six equid taxa (including Cormohipparion emsliei) from the Bone Valley

deposits of central Florida based on analyses of stable carbon isotopes and dental microwear. Both dental microwear and stable carbon isotopes indicate that *Cormohipparion emsliei* was a mixed-feeder. *Hipparion* and *Onohippidium* are not mentioned in MacFadden et al. (1999) and it is unclear what data MacFadden (2005) used to infer their diets. In any case, the dietary assignments for *Cormohipparion*, *Hipparion*, and *Onohippidium* need to be reassessed using proper morphological traits.

According to Table S1, most of the taxa in the study (13 out of 30) were assigned to a dietary category based on the data reported in Janis et al. (2004). Janis et al. (2004) used relative tooth crown height to separate ungulates with a potentially browsing diet from ungulates with other dietary strategies. All low-crowned (brachydont and submesodont; hypsodonty index ≤ 2.5) taxa in that study were assumed to be browsers or animals with a predominantly browsing diet (Janis et al. 2004). Non-brachydont taxa (hypsodonty index > 2.5 ; mesodont and hypsodont taxa) were assumed to have been potentially grazing and/or mixed feeding. Janis et al. (2004) did not attempt to separate potentially grazing taxa from potentially mixed feeding taxa, but rather referred to those taxa collectively as non-brachydonts. Therefore, it is unclear to me how the authors assigned non-brachydont taxa to grazing and mixed-feeding dietary categories. This should be clarified in the manuscript.

A more important issue is that the morphologically-inferred diets presented by the authors were not obtained in a consistent way. Premaxillary shape and other craniodental traits were used in some taxa, whereas the degree of hypsodonty or changes in enamel structure were used in other taxa. This is a concern because these traits can be discordant. For example, *Antilocapra* and *Megatylopus* show discordant dietary reconstructions when looking at their degree of hypsodonty and premaxillary shape. *Antilocapra* is a highly hypsodont ungulate and *Megatylopus* is a mesodont ungulate (Dompiere and Churcher 1996; Janis et al. 2000; Janis et al. 2004; Semprebon et al. 2019); therefore, on the basis of this trait alone they could be interpreted as mixed-feeders or grazers. In contrast, *Antilocapra* and *Megatylopus* have narrow and pointed premaxillaries, which indicate a browsing or predominantly browsing diet (Dompiere and Churcher 1996). One of the factors that very likely contributes to the discordance between these morphological traits is that tooth crown height is known to correlate not only with diet, but also to the degree of openness of the habitat (Janis 1988; Janis et al. 2002; Damuth and Janis, 2011). Therefore, hypsodonty is not necessarily an adaptation for grazing or mixed-feeding, but rather an adaptation to sustain high levels of dental wear, which can arise as a result of eating abrasive plants (such as grasses) and/or ingesting significant amounts of dust and grit (Janis and Fortelius 1988; Damuth and Janis, 2011). Some researchers have even suggested that the development of hypsodonty was primarily driven by an increased exposure to exogenous grit (Semprebon et al. 2019). Relative to other craniodental traits, hypsodonty has been found to be problematic in determining the diet of some ungulates (e.g., Semprebon et al. 2004). With these considerations in mind, if the authors want to use the degree of hypsodonty as a morphological trait that can be consistently evaluated across all taxa in their study, my recommendation is for them to substitute the notion of morphologically-inferred diets and replace it with categories of hypsodonty (i.e., brachydont, submesodont, mesodont, and hypsodont). This will reveal that all brachydont taxa are “locked” as browsers, except for *Platygonus* and *Mylohyus* which extend into the mixed C3-C4 region, possibly as a result of an omnivorous diet. Many hypsodont taxa have $\delta^{13}\text{C}$ values that extend well into the C4 region, but are not precluded from consuming C3 plants. Other hypsodont taxa (e.g., *Antilocapra*, *Capromeryx*, and *Stockoceros*) virtually consume only C3 plants, potentially indicating a uniquely or mostly browsing diet. Some mesodont taxa (e.g., *Hemiauchenia*) span the full C3-C4 spectrum, whereas other mesodont taxa (e.g., *Megatylopus*) have only a C3 diet.

References

Damuth, J. and C. M. Janis. 2011. On the relationship between hypsodonty and feeding ecology in ungulate mammals, and its utility in palaeoecology. *Biological Reviews*. 86 733–758.

Dompierre, H. and C. S. Churcher. 1996. Premaxillary shape as an indicator of the diet of seven extinct late Cenozoic new world camels. *Journal of Vertebrate Paleontology*. 16: 141–148.

Janis, C. M. 1988. An estimation of tooth volume and hypsodonty indices in ungulate mammals, and the correlation of these factors with dietary preferences. In: Russell, D.E., Santoro, J.-P., Sigogneau-Russell, D. (Eds.), *Teeth Revisited: Proceedings of the VIIth International Symposium on Dental Morphology*, Paris, 1986. *Mémoires de Musée d'Histoire Naturelle, Paris, series C*, Paris, France, pp. 367–387.

Janis, C. M. and M. Fortelius. 1988. On the means whereby mammals achieve increased functional durability of their dentitions, with special reference to limiting factors. *Biological Reviews*. 63: 197–230.

Janis, C. M., J. Damuth, and J. M. Theodor. 2000. Miocene ungulates and terrestrial primary productivity: Where have all the browsers gone? *Proceedings of the National Academy of Sciences of the United States of America*. 97: 7899–7904.

Janis, C. M., J. Damuth, and J. M. Theodor. 2002. The origins and evolution of the North American grassland biome: the story from the hoofed mammals. *Palaeogeography, Palaeoclimatology, Palaeoecology*. 177: 183–198.

Janis, C. M., J. Damuth, and J. M. Theodor. 2004. The species richness of Miocene browsers, and implications for habitat type and primary productivity in the North American grassland biome. *Palaeogeography, Palaeoclimatology, Palaeoecology*. 207: 371–398.

MacFadden, B. J. 2005. Fossil horses – Evidence for evolution. *Science*. 307: 1728–1730.

MacFadden, B. J., N. Solounias, and T. E. Cerling. 1999. Ancient diets, ecology, and extinction of 5-million-year-old horses from Florida. *Science*. 283: 824–827.

Semprebon, G. M. and F. Rivals. 2007. Was grass more prevalent in the pronghorn past? An assessment of the dietary adaptations of Miocene to Recent Antilocapridae (Mammalia: Artiodactyla). *Palaeogeography, Palaeoclimatology, Palaeoecology*. 253: 332–347.

Semprebon, G. M., F. Rivals, and C. M. Janis. 2019. The role of grass versus exogenous abrasives in the paleodietary patterns of North American ungulates. *Frontiers in Ecology and Evolution*. 7: Article 65.

Stable isotope analysis

The main assumption that the authors make in their study is that $\delta^{13}\text{C}$ values can reliably distinguish between animals that were grazing, animals that were browsing, and animals that were mixed feeding. A C3 diet is considered to be indicative of browsing, a C4 diet indicative of grazing, and an isotopically mixed diet indicative of mixed feeding. However, this is an assumption that may not hold true across all localities that the authors studied. The authors indicate that restricting their analysis to localities found at lower latitudes (<37° latitude) mitigates the confounding effects of C3 grasses. However, not only is there a latitudinal gradient in C3-C4 grass abundance, but there is also evidence for a general longitudinal gradient, at least during the late Pleistocene in the American Southwest (Connin et al. 1998). C4 grasses appear to decrease in abundance from east to west (Connin et al. 1998). Other studies also suggest decreased C4 grass abundance in different areas of the American Southwest relative to other geographic regions during the Last Glacial Maximum, the mid-Holocene, and the recent past (Cotton et al. 2016). The interaction between precipitation patterns, temperature, and topography could account for the patterns in C3-C4 grass abundance.

I am not sure how the authors can account for the potential longitudinal gradient in C3-C4 grass abundance, other than by geographically restricting their analysis even more. The authors can

explore this possibility. At the very least they need to be clear about their assumptions and models. Therefore, my recommendation to the authors is that they clearly layout their assumptions and acknowledge any potential factors that may confound their results and interpretations. The authors can take a look at MacFadden (2008, section 6.1); I think he does a good job of laying out the assumptions and the model he used to interpret the stable carbon isotope values in his study.

It is also important to point out that the study by MacFadden et al. (1999) provides evidence to suggest that some Hemphillian horses from Florida were grazing on both C3 and C4 grasses. MacFadden et al. (1999) conducted analyses of stable carbon isotopes and dental microwear for six species of horses from the Bone Valley deposits of central Florida. Two of these taxa (*Pseudhipparion simpsoni* and *Nannippus minor*) had dental microwear patterns typical of grazing animals, whereas their stable carbon isotope values suggested a primarily isotopically mixed diet. The results of these analyses potentially indicate the presence of C3 grasses in central Florida during the late Hemphillian.

On a separate note, I realized that the authors did not include the stable carbon isotope data reported by MacFadden et al. (1999) and MacFadden (2008). The authors should incorporate those data in their study, given that one of the objectives of the paper is to provide a synthesis of the stable isotope data of North American ungulates over the past 7 million years. The studies by MacFadden et al. (1999) and MacFadden (2008) provide stable carbon isotope data for two taxa that are omitted in the manuscript by Pardi and DeSantis (*Astrohippus* and *Nannippus*) in addition to data for four taxa that are included in the manuscript.

References

Connin, S. L., J. Betancourt, and J. Quade. 1998. Late Pleistocene C4 plant dominance and summer rainfall in the Southwestern United States from isotopic study of herbivore teeth. *Quaternary Research*. 50: 179–193.

MacFadden, B. J. 2008. Geographic variation in diets of ancient populations of 5-million-year-old (early Pliocene) horses from southern North America. *Palaeogeography, Palaeoclimatology, Palaeoecology*. 266: 83–94.

MacFadden, B. J., N. Solounias, and T. E. Cerling. 1999. Ancient diets, ecology, and extinction of 5-million-year-old horses from Florida. *Science*. 283: 824–827.

Other general comments

I think it would be very useful if the authors could add maps indicating the geographic location of the fossil localities for each time interval they studied. This will make it easier for the reader to visualize the geographic and temporal distribution of the fossil sites included in their study. The maps can be included in the Supplementary Information.

Review form: Reviewer 2

Recommendation

Major revision is needed (please make suggestions in comments)

Scientific importance: Is the manuscript an original and important contribution to its field?

Marginal

General interest: Is the paper of sufficient general interest?

Marginal

Quality of the paper: Is the overall quality of the paper suitable?

Marginal

Is the length of the paper justified?

Yes

Should the paper be seen by a specialist statistical reviewer?

No

Do you have any concerns about statistical analyses in this paper? If so, please specify them explicitly in your report.

No

It is a condition of publication that authors make their supporting data, code and materials available - either as supplementary material or hosted in an external repository. Please rate, if applicable, the supporting data on the following criteria.

Is it accessible?

Yes

Is it clear?

Yes

Is it adequate?

Yes

Do you have any ethical concerns with this paper?

No

Comments to the Author

The present manuscript is an analysis of the congruence between morphological and stable isotopic metrics for paleodiet. The authors compile dietary classifications based on morphology from the literature as well as published stable carbon isotope values from carbonates. Briefly, they find that morphology and reconstructed diet based on $\delta^{13}\text{C}$ values are not always congruent. In particular, species reconstructed as browsers using morphology are more commonly reconstructed as C3 browsers using isotopes than species classified as grazers are reconstructed as C4 grazers. They also show that species at individual sites tend to show lower dietary variation than that of the entire species to which they belong. Furthermore, the dietary breadths of species at single localities are not correlated with estimates of plant abundance, suggesting that competition and site-specific specialization may be at play. The authors suggest that the evolution of grazing "adaptations" such as hypsodonty enable ungulate species to take advantage of a larger range of food stuffs while browsers, by virtue of their brachydont tooth condition, are more dietarily limited. The authors therefore recommend a revision of the categorical terms used to describe dietary categorizations based on morphology.

Overall, the manuscript is well written and the arguments are easy to follow. There are, however, some methodological issues, as outlined below.

The first is that there is a fundamental mismatch between the temporal scales at which morphology and stable isotopes are used to define diet. Changes in morphology represent hundreds to millions of years of evolution. In contrast, stable isotopes represent days to years of an animals life. In the case of a single serial sample from a molar, the $\delta^{13}\text{C}$ values could represent a few days of life (depending on mineralization and maturation rates) while, depending on the taxon, a bulk sample could represent months to years. As a result, we do not necessarily expect 100% congruence of diet reconstructions from morphology and stable isotopes. In the context of the present manuscript, however, the authors seem to implicitly assume that

reconstructions from stable carbon isotopes are representative of lifetime diet, which they demonstrably are not. There is explicit recognition in the stable isotope literature of short term variations in diet that could explain some of the patterns reported in the present manuscript. Morphological and stable isotopic methods are testing different hypotheses relating to diet.

Consideration of the differences in temporal scale are important because this is why people see a deer (*Odocoileus*) eat a dead bird and conclude that they are omnivores or see a deer eat grass and conclude they are grazers, when they are in fact, on average, browsers.

Relevant citation:

Davis, M., and S. Pineda-Munoz. 2016. The temporal scale of diet and dietary proxies. *Ecology and Evolution* 6:1883-1897.

To this end, mixing of brachydont and hypsodont taxa, whose teeth record highly variable amounts of time, could incidentally produce the entire result presented in the manuscript. A single molar of a horse or bison represents more than a year of bulk diet. According to Hoppe et al. some individual horse teeth may record diet over 2.8 years of the animal's life. In contrast, the teeth of brachydont species (deer, for example) mineralize and mature over < 1 year of the animal's life. In this case, the authors are comparing apples and oranges. The observation of greater dietary specialization for browsers is exactly what one might expect simply due to a temporal mismatch. As a result, I see no real way to differentiate dietary specialization from the simple fact that teeth from browser-adapted and grazer-adapted species represent different amounts of the animal's lifetime and thus different temporal scales over which their diets are recorded.

Furthermore, C3 plants have more variable stable carbon isotope ratios. According to Cerling et al. the standard deviation for C3 plants is ~2.7 ‰ whereas for C4 plants it is 1.1‰. So, ability to make comparisons of similar temporal scale for brachydont and hypsodont taxa might result in findings different from those presented in the manuscript. This could be done by looking at comparable amounts of enamel for brachydont species and hypsodont species for which they possess data from serial sampling.

Relevant citations:

Passey, B. H., and T. E. Cerling. 2002. Tooth enamel mineralization in ungulates: Implications for recovering a primary isotopic time-series. *Geochimica et Cosmochimica Acta* 66:3225-3234.

Hoppe, K. A., S. M. Stoverb, J. R. Pascoec, and R. Amundson. 2004. Tooth enamel biomineralization in extant horses: implications for isotopic microsampling. *Palaeogeography, Palaeoclimatology, Palaeoecology* 206:355-365.

Rivera-Araya, Maria, and Suzanne Pilaar Birch. "Stable isotope signatures in white-tailed deer as a seasonal paleoenvironmental proxy: A case study from Georgia, United States." *Palaeogeography, Palaeoclimatology, Palaeoecology* 505 (2018): 53-62.

Cerling, T. E., J. R. Ehleringer, and J. M. Harris. 1998. Carbon Dioxide Starvation, the Development of C 4 Ecosystems, and Mammalian Evolution [and Discussion]. *Philosophical Transactions: Biological Sciences, Vegetation-Climate-Atmosphere Interactions: Past, Present, and Future* 353.

A couple of other important points are missed by the authors. The first is that hypsodont taxa (particularly horses because they are hindgut fermenters) require that their teeth be worn down by the consumption of abrasives (whether this comes from grass or exogenous abrasives is a point of debate, although people seem to have settled on the latter). If not worn down, the teeth will not wear evenly and individuals will be unable to eat (a condition referred to as wave mouth). This is precisely why veterinarians (and farriers) file horse teeth. As such, horses cannot subsist for long periods of time on non-abrasive foods. Thus, the above mentioned mismatch in temporal scale becomes even more important to consider. In the same vein, the paper lacks any discussion of the contribution of abrasives. Perhaps consumption of exogenous abrasives by

grazing-adapted species ameliorated filled the tooth wear requirement.

Relevant citations:

Jardine, P. E., C. M. Janis, S. Sahney, and M. J. Benton. 2012. Grit not grass: Concordant patterns of early origin of hypsodonty in Great Plains ungulates and Glires. *Palaeogeography, Palaeoclimatology, Palaeoecology* 365-366:1-10.

Hoffman, J. M., D. Fraser, and M. T. Clementz. 2015. Controlled feeding trials with ungulates: a new application of in vivo dental molding to assess the abrasive factors of microwear. *The Journal of Experimental Biology* 218:1538-1547.

The authors have attempted to exclude the possibility of C3 grazing by limiting their samples to below 37 degrees of latitude based on the fact that a predictable gradient in C3/C4 grass abundance is present in modern day North America. They, however, have not presented any evidence to corroborate their assumption that the same was true during the latest Miocene and Pliocene. In fact, they present no climate reconstruction or similar that might strengthen their argument. The work of David Fox on stable carbon isotopes from pedogenic carbonates suggests that the modern C3/C4 latitudinal gradient evolved as late as the Pliocene (some of the sites do fall near or right on 37 degrees of latitude, depending on the map projection). In the below cited paper, Fox et al. suggest that the first appearance of ecosystems having >70% C4 plant biomass is observed only 1.3 Ma. But, the assumption that 37 degrees represents a good cut off and a means to reject C3 grazing may, in fact, not be accurate for part of the time period covered by the present study. Even today, ecosystems below 37 degrees of latitude can be between 70 and 100% C4.

Relevant citation:

Fox, D. L., J. G. Honey, R. A. Martin, and P. Peláez-Campomanes. 2012. Pedogenic carbonate stable isotope record of environmental change during the Neogene in the southern Great Plains, southwest Kansas, USA: oxygen isotopes and paleoclimate during the evolution of C4-dominated grasslands. *Geological Society of America Bulletin* 124:431-443.

The authors use a metric for the availability of C3 and C4 resources (the minimum for browsers and maximum for grazers) but do not provide a citation. Does this work in assemblages of mammals where the composition of the ecosystem is known? In mixed, mid-latitude C3/C4 ecosystems do grazers always show a mixed C3/C4 signal? I could see this being true, given that grazers don't necessarily differentiate between C3 and C4 grasses. But it could also be untrue if there is spatial segregation of species with different diets (e.g. grazers feeding in warmer, open areas with more C4 as opposed to the cool understory with more C3). I primarily want to see more justification for their metric. Perhaps this could come in the form of paleoecological reconstructions of the associated plant ecosystems.

There is an implicit assumption that if individual animals are found at sites below 37 degrees of latitude that they never migrated etc. to more northern parts of the continental USA. Some modern equids and ruminants do undergo relatively long-distance seasonal movements. There is no reason to assume that Miocene-Pleistocene ungulates did not. Moreover, Bison are included in their sample and are well known for their north-south migrations in North America. So, though the authors interpret a mixed C3/C4 isotopic signal as representing a mixed browsing/grazing signal, I do not believe they have the power to reject C3 grazing. The associated oxygen isotope data may, however, provide some illumination, though the authors would have to work to differentiate seasonal and migration effects.

Relevant citation:

Larson, R. M., L. C. Todd, E. F. Kelly, and J. M. Welker. 2001. Carbon Stable Isotope Analysis of Bison Dentition. *Great Plains Research: A Journal of Natural and Social Sciences* 11:25-64.

The authors test for differences in the range of isotopic values between individuals at local sites to the entire range of variation for the species (within a time bin, I believe). There is a mathematical issue with this approach because they have not, to my understanding of the paper, demonstrated that localized patterns are differentiable from a random draw from the entire "population" or set

of samples. In fact, they clearly demonstrate what I would expect simply from random sampling that variation at the entire sample scale is higher than at the local scale. Differentiating random effects from localized specialization could be achieved by using something like Cohen's D where observed values are compared to a set of randomizations. If, local values fall outside the range of the randomized values, the authors would have a stronger argument for localized dietary specialization.

A minor point of contention involves the author's statement that stable isotope values are phylogenetically independent (without relevant citation) (line 65). This is demonstrably untrue. At the very least, I would expect a division between the artiodactyls and perissodactyls based on their dietary physiology, which results in different degrees of carbon and nitrogen fractionation. I think it is naïve to assume that reconstructions of diet from stable isotopes are "taxon free." This requires the assumption that physiology is invariant across mammalian taxa. We know this very likely isn't true. However, I reiterate that this is a minor point because their analyses probably do not require phylogenetic correction.

Relevant citations:

Hedges, Robert EM. "On bone collagen – apatite-carbonate isotopic relationships." *International Journal of Osteoarchaeology* 13.1-2 (2003): 66-79.

Codron, Daryl, et al. "Within trophic level shifts in collagen-carbonate stable carbon isotope spacing are propagated by diet and digestive physiology in large mammal herbivores." *Ecology and evolution* 8.8 (2018): 3983-3995.

Decision letter (RSPB-2020-0945.R0)

22-Jun-2020

Dear Dr Pardi:

I am writing to inform you that your manuscript RSPB-2020-0945 entitled "Dietary plasticity of North American herbivores: A synthesis of stable isotope data over the past 7 million years" has, in its current form, been rejected for publication in *Proceedings B*.

This action has been taken on the advice of referees, who have recommended that substantial revisions are necessary. With this in mind we would be happy to consider a resubmission, provided the comments of the referees are fully addressed. However please note that this is not a provisional acceptance.

Please note that this decision may (or may not) have taken into account confidential comments.

In your revision process, please take a second look at how open your science is; our policy is that **ALL** (maximally inclusive) data involved with the study should be made openly accessible, fully enabling re-use, replication and transparency-- see:

<https://royalsociety.org/journals/ethics-policies/data-sharing-mining/>

Insufficient sharing of data can delay or even cause rejection of a paper.

Full data and code/scripts to enable reuse/replication/repurposing are what this policy intends.

Sincerely,

Dr John Hutchinson, Editor

Associate Editor

Board Member: 1

Comments to Author:

To the Authors, you will find attached to this correspondence, the comments of two referees both of whom are recommending major revisions to your manuscript. I find I am in agreement with the comments and criticisms of both referees, and further that they are well intended, clear and constructive. The comments and criticisms converge on similar points regarding your assessment of C3 and C4 plants, carbon isotopes, methods, the relevant mammal groups and your assessment of physiology, etc. I do not see that these criticisms are substantive, though there is a requirement for manuscript edits, additional citations, and some modest reassessment of analyses and methods.

Please undertake the requested revisions or rebut those requests if you feel it is necessary. For all comments and criticisms, those you agree with and thus you are choosing to respond to as rebuttals, please create and submit a fully detailed cover/rebuttal letter. Full consideration of your responses will be given in determining the next steps regarding your manuscript submission.

Reviewer(s)' Comments to Author:

Referee: 1

Comments to the Author(s)

Pardi and DeSantis have presented a very interesting study that adds significantly to our understanding of the feeding ecology of Neogene North American herbivores. I think the study is of appropriate relevance and broad interest to be considered for publication in the Proceedings of the Royal Society B. However, I have several comments and concerns that I would like the authors to consider. My concerns and comments are detailed below.

Morphologically-inferred diets

There are inconsistencies in the morphological traits that the authors cite as evidence to morphologically infer the diet of the taxa in their study. Based on the references provided in Table S1, some of the taxa are assigned to a dietary category based on evidence from dental microwear (Capromeryx, Cormohipparion, Stockoceros, and Tetrameryx), degree of hypsodonty (13 out of the 30 taxa studied), premaxillary shape (all camelids), changes in enamel arrangement and/or structure (Cuvieronius and Mammuthus), and multivariate assessment of craniodental features (Pediomeryx). In my opinion, the application of dental microwear and degree of hypsodonty to morphologically infer the diet of the taxa in the study are the most problematic and my concerns are expressed below.

The authors assigned the extinct antilocaprids (*Capromeryx*, *Stockoceros*, and *Tetrameryx*) to a dietary category based on the study by Semprebon and Rivals (2007). Semprebon and Rivals (2007) made dietary inferences for those taxa based on their analysis of dental microwear. As Pardi and DeSantis point out in the Introduction of their manuscript, dental microwear and stable isotopes provide insights into “what an individual animal consumed at a specific place and moment in time, as opposed to its potential diet as inferred from morphological features...” (Pardi and DeSantis page 4). Therefore, the authors should reassess the dietary assignment of the extinct antilocaprids using proper morphological traits.

For the equids *Cormohipparion*, *Hipparion*, and *Onohippidium* the authors cite MacFadden (2005) as the source of their morphologically assigned diet. MacFadden (2005) presents a figure that summarizes the phylogeny, geographic distribution, diet, and body sizes of the family Equidae at the generic level. In this figure, *Cormohipparion* is identified as a mixed-feeder, *Hipparion* as a grazer, and *Onohippidium* as a mixed-feeder. MacFadden (2005) does not explicitly state what kind of data was used to infer the diet of the different genera shown in the figure. However, in the text he mentions that some hypsodont equids were grazers and others mixed-feeders, citing the study by MacFadden et al. (1999). The study by MacFadden et al. (1999) inferred the diets of six equid taxa (including *Cormohipparion emsleyi*) from the Bone Valley deposits of central Florida based on analyses of stable carbon isotopes and dental microwear. Both dental microwear and stable carbon isotopes indicate that *Cormohipparion emsleyi* was a mixed-feeder. *Hipparion* and *Onohippidium* are not mentioned in MacFadden et al. (1999) and it is unclear what data MacFadden (2005) used to infer their diets. In any case, the dietary assignments for *Cormohipparion*, *Hipparion*, and *Onohippidium* need to be reassessed using proper morphological traits.

According to Table S1, most of the taxa in the study (13 out of 30) were assigned to a dietary category based on the data reported in Janis et al. (2004). Janis et al. (2004) used relative tooth crown height to separate ungulates with a potentially browsing diet from ungulates with other dietary strategies. All low-crowned (brachydont and submesodont; hypsodonty index ≤ 2.5) taxa in that study were assumed to be browsers or animals with a predominantly browsing diet (Janis et al. 2004). Non-brachydont taxa (hypsodonty index > 2.5 ; mesodont and hypsodont taxa) were assumed to have been potentially grazing and/or mixed feeding. Janis et al. (2004) did not attempt to separate potentially grazing taxa from potentially mixed feeding taxa, but rather referred to those taxa collectively as non-brachydonts. Therefore, it is unclear to me how the authors assigned non-brachydont taxa to grazing and mixed-feeding dietary categories. This should be clarified in the manuscript.

A more important issue is that the morphologically-inferred diets presented by the authors were not obtained in a consistent way. Premaxillary shape and other craniodental traits were used in some taxa, whereas the degree of hypsodonty or changes in enamel structure were used in other taxa. This is a concern because these traits can be discordant. For example, *Antilocapra* and *Megatylopus* show discordant dietary reconstructions when looking at their degree of hypsodonty and premaxillary shape. *Antilocapra* is a highly hypsodont ungulate and *Megatylopus* is a mesodont ungulate (Dompiere and Churcher 1996; Janis et al. 2000; Janis et al. 2004; Semprebon et al. 2019); therefore, on the basis of this trait alone they could be interpreted as mixed-feeders or grazers. In contrast, *Antilocapra* and *Megatylopus* have narrow and pointed premaxillaries, which indicate a browsing or predominantly browsing diet (Dompiere and Churcher 1996). One of the factors that very likely contributes to the discordance between these morphological traits is that tooth crown height is known to correlate not only with diet, but also to the degree of openness of the habitat (Janis 1988; Janis et al. 2002; Damuth and Janis, 2011). Therefore, hypsodonty is not necessarily an adaptation for grazing or mixed-feeding, but rather an adaptation to sustain high levels of dental wear, which can arise as a result of eating abrasive plants (such as grasses) and/or ingesting significant amounts of dust and grit (Janis and Fortelius 1988; Damuth and Janis, 2011). Some researchers have even suggested that the development of hypsodonty was primarily driven by an increased exposure to exogenous grit (Semprebon et al.

2019). Relative to other craniodental traits, hypsodonty has been found to be problematic in determining the diet of some ungulates (e.g., Semprebon et al. 2004). With these considerations in mind, if the authors want to use the degree of hypsodonty as a morphological trait that can be consistently evaluated across all taxa in their study, my recommendation is for them to substitute the notion of morphologically-inferred diets and replace it with categories of hypsodonty (i.e., brachydont, submesodont, mesodont, and hypsodont). This will reveal that all brachydont taxa are “locked” as browsers, except for *Platygonus* and *Mylohyus* which extend into the mixed C3-C4 region, possibly as a result of an omnivorous diet. Many hypsodont taxa have $\delta^{13}\text{C}$ values that extend well into the C4 region, but are not precluded from consuming C3 plants. Other hypsodont taxa (e.g., *Antilocapra*, *Capromeryx*, and *Stockoceros*) virtually consume only C3 plants, potentially indicating a uniquely or mostly browsing diet. Some mesodont taxa (e.g., *Hemiauchenia*) span the full C3-C4 spectrum, whereas other mesodont taxa (e.g., *Megatylopus*) have only a C3 diet.

References

- Damuth, J. and C. M. Janis. 2011. On the relationship between hypsodonty and feeding ecology in ungulate mammals, and its utility in palaeoecology. *Biological Reviews*. 86: 733–758.
- Dompierre, H. and C. S. Churcher. 1996. Premaxillary shape as an indicator of the diet of seven extinct late Cenozoic new world camels. *Journal of Vertebrate Paleontology*. 16: 141–148.
- Janis, C. M. 1988. An estimation of tooth volume and hypsodonty indices in ungulate mammals, and the correlation of these factors with dietary preferences. In: Russell, D.E., Santoro, J.-P., Sigogneau-Russell, D. (Eds.), *Teeth Revisited: Proceedings of the VIIth International Symposium on Dental Morphology*, Paris, 1986. Mémoires de Musée d'Histoire Naturelle, Paris, series C, Paris, France, pp. 367–387.
- Janis, C. M. and M. Fortelius. 1988. On the means whereby mammals achieve increased functional durability of their dentitions, with special reference to limiting factors. *Biological Reviews*. 63: 197–230.
- Janis, C. M., J. Damuth, and J. M. Theodor. 2000. Miocene ungulates and terrestrial primary productivity: Where have all the browsers gone? *Proceedings of the National Academy of Sciences of the United States of America*. 97: 7899–7904.
- Janis, C. M., J. Damuth, and J. M. Theodor. 2002. The origins and evolution of the North American grassland biome: the story from the hoofed mammals. *Palaeogeography, Palaeoclimatology, Palaeoecology*. 177: 183–198.
- Janis, C. M., J. Damuth, and J. M. Theodor. 2004. The species richness of Miocene browsers, and implications for habitat type and primary productivity in the North American grassland biome. *Palaeogeography, Palaeoclimatology, Palaeoecology*. 207: 371–398.
- MacFadden, B. J. 2005. Fossil horses – Evidence for evolution. *Science*. 307: 1728–1730.
- MacFadden, B. J., N. Solounias, and T. E. Cerling. 1999. Ancient diets, ecology, and extinction of 5-million-year-old horses from Florida. *Science*. 283: 824–827.
- Semprebon, G. M. and F. Rivals. 2007. Was grass more prevalent in the pronghorn past? An assessment of the dietary adaptations of Miocene to Recent Antilocapridae (Mammalia: Artiodactyla). *Palaeogeography, Palaeoclimatology, Palaeoecology*. 253: 332–347.
- Semprebon, G. M., F. Rivals, and C. M. Janis. 2019. The role of grass versus exogenous abrasives in the paleodietary patterns of North American ungulates. *Frontiers in Ecology and Evolution*. 7: Article 65.

Stable isotope analysis

The main assumption that the authors make in their study is that $\delta^{13}\text{C}$ values can reliably distinguish between animals that were grazing, animals that were browsing, and animals that were mixed feeding. A C3 diet is considered to be indicative of browsing, a C4 diet indicative of grazing, and an isotopically mixed diet indicative of mixed feeding. However, this is an assumption that may not hold true across all localities that the authors studied. The authors indicate that restricting their analysis to localities found at lower latitudes (<37° latitude) mitigates the confounding effects of C3 grasses. However, not only is there a latitudinal gradient in C3-C4 grass abundance, but there is also evidence for a general longitudinal gradient, at least during the late Pleistocene in the American Southwest (Connin et al. 1998). C4 grasses appear to decrease in abundance from east to west (Connin et al. 1998). Other studies also suggest decreased C4 grass abundance in different areas of the American Southwest relative to other geographic regions during the Last Glacial Maximum, the mid-Holocene, and the recent past (Cotton et al. 2016). The interaction between precipitation patterns, temperature, and topography could account for the patterns in C3-C4 grass abundance.

I am not sure how the authors can account for the potential longitudinal gradient in C3-C4 grass abundance, other than by geographically restricting their analysis even more. The authors can explore this possibility. At the very least they need to be clear about their assumptions and models. Therefore, my recommendation to the authors is that they clearly layout their assumptions and acknowledge any potential factors that may confound their results and interpretations. The authors can take a look at MacFadden (2008, section 6.1); I think he does a good job of laying out the assumptions and the model he used to interpret the stable carbon isotope values in his study.

It is also important to point out that the study by MacFadden et al. (1999) provides evidence to suggest that some Hemphillian horses from Florida were grazing on both C3 and C4 grasses. MacFadden et al. (1999) conducted analyses of stable carbon isotopes and dental microwear for six species of horses from the Bone Valley deposits of central Florida. Two of these taxa (*Pseudhipparion simpsoni* and *Nannippus minor*) had dental microwear patterns typical of grazing animals, whereas their stable carbon isotope values suggested a primarily isotopically mixed diet. The results of these analyses potentially indicate the presence of C3 grasses in central Florida during the late Hemphillian.

On a separate note, I realized that the authors did not include the stable carbon isotope data reported by MacFadden et al. (1999) and MacFadden (2008). The authors should incorporate those data in their study, given that one of the objectives of the paper is to provide a synthesis of the stable isotope data of North American ungulates over the past 7 million years. The studies by MacFadden et al. (1999) and MacFadden (2008) provide stable carbon isotope data for two taxa that are omitted in the manuscript by Pardi and DeSantis (*Astrohippus* and *Nannippus*) in addition to data for four taxa that are included in the manuscript.

References

Connin, S. L., J. Betancourt, and J. Quade. 1998. Late Pleistocene C4 plant dominance and summer rainfall in the Southwestern United States from isotopic study of herbivore teeth. *Quaternary Research*. 50: 179–193.

MacFadden, B. J. 2008. Geographic variation in diets of ancient populations of 5-million-year-old (early Pliocene) horses from southern North America. *Palaeogeography, Palaeoclimatology, Palaeoecology*. 266: 83–94.

MacFadden, B. J., N. Solounias, and T. E. Cerling. 1999. Ancient diets, ecology, and extinction of 5-million-year-old horses from Florida. *Science*. 283: 824–827.

Other general comments

I think it would be very useful if the authors could add maps indicating the geographic location of the fossil localities for each time interval they studied. This will make it easier for the reader to visualize the geographic and temporal distribution of the fossil sites included in their study. The maps can be included in the Supplementary Information.

Referee: 2

Comments to the Author(s)

The present manuscript is an analysis of the congruence between morphological and stable isotopic metrics for paleodiet. The authors compile dietary classifications based on morphology from the literature as well as published stable carbon isotope values from carbonates. Briefly, they find that morphology and reconstructed diet based on $\delta^{13}\text{C}$ values are not always congruent. In particular, species reconstructed as browsers using morphology are more commonly reconstructed as C3 browsers using isotopes than species classified as grazers are reconstructed as C4 grazers. They also show that species at individual sites tend to show lower dietary variation than that of the entire species to which they belong. Furthermore, the dietary breadths of species at single localities are not correlated with estimates of plant abundance, suggesting that competition and site-specific specialization may be at play. The authors suggest that the evolution of grazing "adaptations" such as hypsodonty enable ungulate species to take advantage of a larger range of food stuffs while browsers, by virtue of their brachydont tooth condition, are more dietarily limited. The authors therefore recommend a revision of the categorical terms used to describe dietary categorizations based on morphology.

Overall, the manuscript is well written and the arguments are easy to follow. There are, however, some methodological issues, as outlined below.

The first is that there is a fundamental mismatch between the temporal scales at which morphology and stable isotopes are used to define diet. Changes in morphology represent hundreds to millions of years of evolution. In contrast, stable isotopes represent days to years of an animal's life. In the case of a single serial sample from a molar, the $\delta^{13}\text{C}$ values could represent a few days of life (depending on mineralization and maturation rates) while, depending on the taxon, a bulk sample could represent months to years. As a result, we do not necessarily expect 100% congruence of diet reconstructions from morphology and stable isotopes. In the context of the present manuscript, however, the authors seem to implicitly assume that reconstructions from stable carbon isotopes are representative of lifetime diet, which they demonstrably are not. There is explicit recognition in the stable isotope literature of short term variations in diet that could explain some of the patterns reported in the present manuscript. Morphological and stable isotopic methods are testing different hypotheses relating to diet.

Consideration of the differences in temporal scale are important because this is why people see a deer (*Odocoileus*) eat a dead bird and conclude that they are omnivores or see a deer eat grass and conclude they are grazers, when they are in fact, on average, browsers.

Relevant citation:

Davis, M., and S. Pineda-Munoz. 2016. The temporal scale of diet and dietary proxies. *Ecology and Evolution* 6:1883-1897.

To this end, mixing of brachydont and hypsodont taxa, whose teeth record highly variable amounts of time, could incidentally produce the entire result presented in the manuscript. A single molar of a horse or bison represents more than a year of bulk diet. According to Hoppe et al. some individual horse teeth may record diet over 2.8 years of the animal's life. In contrast, the teeth of brachydont species (deer, for example) mineralize and mature over < 1 year of the animal's life. In this case, the authors are comparing apples and oranges. The observation of greater dietary specialization for browsers is exactly what one might expect simply due to a

temporal mismatch. As a result, I see no real way to differentiate dietary specialization from the simple fact that teeth from browser-adapted and grazer-adapted species represent different amounts of the animal's lifetime and thus different temporal scales over which their diets are recorded.

Furthermore, C3 plants have more variable stable carbon isotope ratios. According to Cerling et al. the standard deviation for C3 plants is ~2.7 ‰ whereas for C4 plants it is 1.1‰. So, ability to make comparisons of similar temporal scale for brachydont and hypsodont taxa might result in findings different from those presented in the manuscript. This could be done by looking at comparable amounts of enamel for brachydont species and hypsodont species for which they possess data from serial sampling.

Relevant citations:

Passey, B. H., and T. E. Cerling. 2002. Tooth enamel mineralization in ungulates: Implications for recovering a primary isotopic time-series. *Geochimica et Cosmochimica Acta* 66:3225–3234.

Hoppe, K. A., S. M. Stoverb, J. R. Pascoec, and R. Amundson. 2004. Tooth enamel biomineralization in extant horses: implications for isotopic microsampling. *Palaeogeography, Palaeoclimatology, Palaeoecology* 206:355-365.

Rivera-Araya, Maria, and Suzanne Pilaar Birch. "Stable isotope signatures in white-tailed deer as a seasonal paleoenvironmental proxy: A case study from Georgia, United States."

Palaeogeography, Palaeoclimatology, Palaeoecology 505 (2018): 53-62.

Cerling, T. E., J. R. Ehleringer, and J. M. Harris. 1998. Carbon Dioxide Starvation, the Development of C4 Ecosystems, and Mammalian Evolution [and Discussion]. *Philosophical Transactions: Biological Sciences, Vegetation-Climate-Atmosphere Interactions: Past, Present, and Future* 353.

A couple of other important points are missed by the authors. The first is that hypsodont taxa (particularly horses because they are hindgut fermenters) require that their teeth be worn down by the consumption of abrasives (whether this comes from grass or exogenous abrasives is a point of debate, although people seem to have settled on the latter). If not worn down, the teeth will not wear evenly and individuals will be unable to eat (a condition referred to as wave mouth). This is precisely why veterinarians (and farriers) file horse teeth. As such, horses cannot subsist for long periods of time on non-abrasive foods. Thus, the above mentioned mismatch in temporal scale becomes even more important to consider. In the same vein, the paper lacks any discussion of the contribution of abrasives. Perhaps consumption of exogenous abrasives by grazing-adapted species ameliorated filled the tooth wear requirement.

Relevant citations:

Jardine, P. E., C. M. Janis, S. Sahney, and M. J. Benton. 2012. Grit not grass: Concordant patterns of early origin of hypsodonty in Great Plains ungulates and Glires. *Palaeogeography, Palaeoclimatology, Palaeoecology* 365-366:1-10.

Hoffman, J. M., D. Fraser, and M. T. Clementz. 2015. Controlled feeding trials with ungulates: a new application of in vivo dental molding to assess the abrasive factors of microwear. *The Journal of Experimental Biology* 218:1538-1547.

The authors have attempted to exclude the possibility of C3 grazing by limiting their samples to below 37 degrees of latitude based on the fact that a predictable gradient in C3/C4 grass abundance is present in modern day North America. They, however, have not presented any evidence to corroborate their assumption that the same was true during the latest Miocene and Pliocene. In fact, they present no climate reconstruction or similar that might strengthen their argument. The work of David Fox on stable carbon isotopes from pedogenic carbonates suggests that the modern C3/C4 latitudinal gradient evolved as late as the Pliocene (some of the sites do fall near or right on 37 degrees of latitude, depending on the map projection). In the below cited paper, Fox et al. suggest that the first appearance of ecosystems having >70% C4 plant biomass is observed only 1.3 Ma. But, the assumption that 37 degrees represents a good cut off and a means

to reject C3 grazing may, in fact, not be accurate for part of the time period covered by the present study. Even today, ecosystems below 37 degrees of latitude can be between 70 and 100% C4.

Relevant citation:

Fox, D. L., J. G. Honey, R. A. Martin, and P. Peláez-Campomanes. 2012. Pedogenic carbonate stable isotope record of environmental change during the Neogene in the southern Great Plains, southwest Kansas, USA: oxygen isotopes and paleoclimate during the evolution of C4-dominated grasslands. *Geological Society of America Bulletin* 124:431-443.

The authors use a metric for the availability of C3 and C4 resources (the minimum for browsers and maximum for grazers) but do not provide a citation. Does this work in assemblages of mammals where the composition of the ecosystem is known? In mixed, mid-latitude C3/C4 ecosystems do grazers always show a mixed C3/C4 signal? I could see this being true, given that grazers don't necessarily differentiate between C3 and C4 grasses. But it could also be untrue if there is spatial segregation of species with different diets (e.g. grazers feeding in warmer, open areas with more C4 as opposed to the cool understory with more C3). I primarily want to see more justification for their metric. Perhaps this could come in the form of paleoecological reconstructions of the associated plant ecosystems.

There is an implicit assumption that if individual animals are found at sites below 37 degrees of latitude that they never migrated etc. to more northern parts of the continental USA. Some modern equids and ruminants do undergo relatively long-distance seasonal movements. There is no reason to assume that Miocene-Pleistocene ungulates did not. Moreover, Bison are included in their sample and are well known for their north-south migrations in North America. So, though the authors interpret a mixed C3/C4 isotopic signal as representing a mixed browsing/grazing signal, I do not believe they have the power to reject C3 grazing. The associated oxygen isotope data may, however, provide some illumination, though the authors would have to work to differentiate seasonal and migration effects.

Relevant citation:

Larson, R. M., L. C. Todd, E. F. Kelly, and J. M. Welker. 2001. Carbon Stable Isotope Analysis of Bison Dentition. *Great Plains Research: A Journal of Natural and Social Sciences* 11:25-64.

The authors test for differences in the range of isotopic values between individuals at local sites to the entire range of variation for the species (within a time bin, I believe). There is a mathematical issue with this approach because they have not, to my understanding of the paper, demonstrated that localized patterns are differentiable from a random draw from the entire "population" or set of samples. In fact, they clearly demonstrate what I would expect simply from random sampling that variation at the entire sample scale is higher than at the local scale. Differentiating random effects from localized specialization could be achieved by using something like Cohen's D where observed values are compared to a set of randomizations. If, local values fall outside the range of the randomized values, the authors would have a stronger argument for localized dietary specialization.

A minor point of contention involves the author's statement that stable isotope values are phylogenetically independent (without relevant citation) (line 65). This is demonstrably untrue. At the very least, I would expect a division between the artiodactyls and perissodactyls based on their dietary physiology, which results in different degrees of carbon and nitrogen fractionation. I think it is naïve to assume that reconstructions of diet from stable isotopes are "taxon free." This requires the assumption that physiology is invariant across mammalian taxa. We know this very likely isn't true. However, I reiterate that this is a minor point because their analyses probably do not require phylogenetic correction.

Relevant citations:

Hedges, Robert EM. "On bone collagen – apatite-carbonate isotopic relationships." *International Journal of Osteoarchaeology* 13.1-2 (2003): 66-79.

Codron, Daryl, et al. "Within trophic level shifts in collagen-carbonate stable carbon isotope spacing are propagated by diet and digestive physiology in large mammal herbivores." *Ecology and evolution* 8.8 (2018): 3983-3995.

Author's Response to Decision Letter for (RSPB-2020-0945.R0)

See Appendix A.

RSPB-2021-0121.R0

Review form: Reviewer 1 (Christina Barron-Ortiz)

Recommendation

Accept with minor revision (please list in comments)

Scientific importance: Is the manuscript an original and important contribution to its field?

Excellent

General interest: Is the paper of sufficient general interest?

Good

Quality of the paper: Is the overall quality of the paper suitable?

Excellent

Is the length of the paper justified?

Yes

Should the paper be seen by a specialist statistical reviewer?

No

Do you have any concerns about statistical analyses in this paper? If so, please specify them explicitly in your report.

No

It is a condition of publication that authors make their supporting data, code and materials available - either as supplementary material or hosted in an external repository. Please rate, if applicable, the supporting data on the following criteria.

Is it accessible?

Yes

Is it clear?

Yes

Is it adequate?

Yes

Do you have any ethical concerns with this paper?

No

Comments to the Author

The authors have addressed all of the concerns I indicated on my previous review. However, I still have some minor observations and comments that I would like the authors to consider.

I noticed that on Figure 2 and Figure S8, you have some mammoth specimens identified as Blancan in age. Currently, the Irvingtonian NALMA is defined by the first appearance of *Mammuthus* in North America south of 55° N (Bell et al. 2004). Therefore, by definition, the mammoth specimens you are plotting here cannot be Blancan in age. Please check the age of these specimens and other specimens from that site to make sure they are assigned to the correct time bin. I am sorry I missed this observation on my previous review. The fact that you are now plotting the data by degree of hypsodonty made this error really stick out; *Mammuthus* is the only hypsodont proboscidean in the dataset.

The observation mentioned above made me look at your raw data in more detail. I noticed that specimens from Inglis 1A are assigned to the Irvingtonian, but I believe this site is currently considered latest Blancan in age (Bell et al. 2004). This and the error mentioned above are the only errors I was able to identify, but I am not completely familiar with all of the localities you studied. So, please make sure that you are being consistent when assigning specimens to each time bin.

Bell, C. J., Lundelius, E. L. Jr., Barnosky, A. D., Graham, R. W., Lindsay, E. H., Ruez, D. R. Jr., et al. 2004. "The Blancan, Irvingtonian, and Rancholabrean Mammal Ages," in *Late Cretaceous and Cenozoic Mammals of North America: Biostratigraphy and Geochronology*, ed M. O. Woodburne (New York, NY: Columbia University Press), 232-314. doi: 10.7312/wood13040-009

Page 4, line 53: There is a typo here. Please change "...evolutions..." to "...evolution's..."

Page 4, line 53: To improve clarity, I suggest rephrasing "...examples of an animal's ability to adapt..." to "...examples of the ability of animals to adapt..."

Page 4, line 57: I find this a bit confusing. You state that "...it is also evident that certain forms do not necessitate specific diets...". By forms do you mean morphology, individual animals, or species?

Page 4, line 58: Just to be more precise, I suggest changing "...this allows for the inclusion of grass into their diet..." to "...this allows for the inclusion of grass and other abrasive food items into their diet..."

Page 4, line 65: There is a typo here. Please change "...shaped evolutionary timescales..." to "...shaped in evolutionary timescales..."

Page 5, lines 72 - 73: Here you ask: "...Are grazing taxa specialists, or is grazing a means to broaden the dietary niche?" For "grazing taxa" are you referring to taxa identified as "grazers" based on morphological criteria or to taxa identified as grazers based on the isotope data, according to the criteria you outline in the methodology? If you are referring to taxa identified as "grazers" based on morphological criteria, I suggest substituting "grazing taxa" for hypsodont taxa: Are hypsodont taxa specialists, or is hypsodonty a means to broaden the dietary niche.

Page 7, line 125: There is a typo here. Please change "...were where..." to "...where..."

Page 10, line 193: There is a typo here. Please change "...Perisodactyls..." to "...Perissodactyls..."

Page 11, line 213: Please substitute underlined text with normal text.

Page 17, line 359: Do you mean dietary categorizations based on morphology?

Page 18, line 381: Please change "...grazing megafauna..." to "...grazing-adapted megafauna..."

Page 18, line 389: I suggest replacing "browsers" with "brachydont taxa", so that you refer to a morphological trait typically associated with browsing.

Page 29, line 601: Please change "...the latest record specimens..." to "...the oldest recorded specimens..."

Review form: Reviewer 2

Recommendation

Major revision is needed (please make suggestions in comments)

Scientific importance: Is the manuscript an original and important contribution to its field?

Marginal

General interest: Is the paper of sufficient general interest?

Marginal

Quality of the paper: Is the overall quality of the paper suitable?

Marginal

Is the length of the paper justified?

Yes

Should the paper be seen by a specialist statistical reviewer?

No

Do you have any concerns about statistical analyses in this paper? If so, please specify them explicitly in your report.

No

It is a condition of publication that authors make their supporting data, code and materials available - either as supplementary material or hosted in an external repository. Please rate, if applicable, the supporting data on the following criteria.

Is it accessible?

Yes

Is it clear?

Yes

Is it adequate?

Yes

Do you have any ethical concerns with this paper?

No

Comments to the Author

I have included my comments as a PDF because I wanted to include some figures pulled from published papers. (See Appendix B)

Decision letter (RSPB-2021-0121.R0)

04-Mar-2021

Dear Dr Pardi

I am pleased to inform you that your manuscript RSPB-2021-0121 entitled "Dietary plasticity of North American herbivores: A synthesis of stable isotope data over the past 7 million years" has been accepted for publication in Proceedings B. Congratulations!!

The referee(s) have recommended publication, but also suggest some minor revisions to your manuscript. Therefore, I invite you to respond to the referee(s)' comments and revise your manuscript. Because the schedule for publication is very tight, it is a condition of publication that you submit the revised version of your manuscript within 7 days. If you do not think you will be able to meet this date please let us know.

We obtained one review that had minor recommendations. Another was late and was deemed to be too casual and potentially hostile, so we have omitted it here in our judgement; only the first reviewer's recommendations need to be satisfied.

Sincerely,

Dr John Hutchinson, Editor

Associate Editor

Board Member

Comments to Author:

I am recommending acceptance with minor revisions from a single Reviewer, who has provided a short list of editorial revisions to the manuscript and has also recommended Acceptance with Minor Revisions. I find the manuscript to have addressed, through two rounds of reviews, the requested revisions of two reviewers.

Reviewer(s)' Comments to Author:

Referee: 1

Comments to the Author(s).

The authors have addressed all of the concerns I indicated on my previous review. However, I still have some minor observations and comments that I would like the authors to consider.

I noticed that on Figure 2 and Figure S8, you have some mammoth specimens identified as Blancan in age. Currently, the Irvingtonian NALMA is defined by the first appearance of *Mammuthus* in North America south of 55° N (Bell et al. 2004). Therefore, by definition, the mammoth specimens you are plotting here cannot be Blancan in age. Please check the age of these specimens and other specimens from that site to make sure they are assigned to the correct time bin. I am sorry I missed this observation on my previous review. The fact that you are now plotting the data by degree of hypsodonty made this error really stick out; *Mammuthus* is the only hypsodont proboscidean in the dataset.

The observation mentioned above made me look at your raw data in more detail. I noticed that specimens from Inglis 1A are assigned to the Irvingtonian, but I believe this site is currently considered latest Blancan in age (Bell et al. 2004). This and the error mentioned above are the only errors I was able to identify, but I am not completely familiar with all of the localities you studied. So, please make sure that you are being consistent when assigning specimens to each time bin.

Bell, C. J., Lundelius, E. L. Jr., Barnosky, A. D., Graham, R. W., Lindsay, E. H., Ruez, D. R. Jr., et al. 2004. "The Blancan, Irvingtonian, and Rancholabrean Mammal Ages," in *Late Cretaceous and Cenozoic Mammals of North America: Biostratigraphy and Geochronology*, ed M. O. Woodburne (New York, NY: Columbia University Press), 232-314. doi: 10.7312/wood13040-009

Page 4, line 53: There is a typo here. Please change "...evolutions..." to "...evolution's..."

Page 4, line 53: To improve clarity, I suggest rephrasing "...examples of an animal's ability to adapt..." to "...examples of the ability of animals to adapt..."

Page 4, line 57: I find this a bit confusing. You state that "...it is also evident that certain forms do not necessitate specific diets...". By forms do you mean morphology, individual animals, or species?

Page 4, line 58: Just to be more precise, I suggest changing "...this allows for the inclusion of grass into their diet..." to "...this allows for the inclusion of grass and other abrasive food items into their diet..."

Page 4, line 65: There is a typo here. Please change "...shaped evolutionary timescales..." to "...shaped in evolutionary timescales..."

Page 5, lines 72 - 73: Here you ask: "...Are grazing taxa specialists, or is grazing a means to broaden the dietary niche?" For "grazing taxa" are you referring to taxa identified as "grazers" based on morphological criteria or to taxa identified as grazers based on the isotope data, according to the criteria you outline in the methodology? If you are referring to taxa identified as "grazers" based on morphological criteria, I suggest substituting "grazing taxa" for hypsodont taxa: Are hypsodont taxa specialists, or is hypsodonty a means to broaden the dietary niche.

Page 7, line 125: There is a typo here. Please change "...were where..." to "...where..."

Page 10, line 193: There is a typo here. Please change "...Perisodactyls..." to "...Perissodactyls..."

Page 11, line 213: Please substitute underlined text with normal text.

Page 17, line 359: Do you mean dietary categorizations based on morphology?

Page 18, line 381: Please change "...grazing megafauna..." to "...grazing-adapted megafauna..."

Page 18, line 389: I suggest replacing "browsers" with "brachyodont taxa", so that you refer to a morphological trait typically associated with browsing.

Page 29, line 601: Please change "...the latest record specimens..." to "...the oldest recorded specimens..."

Author's Response to Decision Letter for (RSPB-2021-0121.R0)

See Appendix C.

Decision letter (RSPB-2021-0121.R1)

17-Mar-2021

Dear Dr Pardi

I am pleased to inform you that your manuscript entitled "Dietary plasticity of North American herbivores: A synthesis of stable isotope data over the past 7 million years" has been accepted for publication in Proceedings B.

Data Accessibility section

Open Access

Paper charges

All supplementary materials accompanying an accepted article will be treated as in their final form. They will be published alongside the paper on the journal website and posted on the online

figshare repository. Files on figshare will be made available approximately one week before the accompanying article so that the supplementary material can be attributed a unique DOI.

Sincerely,
Proceedings B
<mailto:proceedingsb@royalsociety.org>

Appendix A

Thank you for the opportunity to address the issues brought forth by the two reviewers who assessed our submission. Reviewer or Board Member comments are provided in *italics* with author comments to editorial staff and reviewers provided in plain text.

Associate Editor

Board Member: 1

Comments to Author:

To the Authors, you will find attached to this correspondence, the comments of two referees both of whom are recommending major revisions to your manuscript. I find I am in agreement with the comments and criticisms of both referees, and further that they are well intended, clear and constructive. The comments and criticisms converge on similar points regarding your assessment of C3 and C4 plants, carbon isotopes, methods, the relevant mammal groups and your assessment of physiology, etc. I do not see that these criticisms are substantive, though there is a requirement for manuscript edits, additional citations, and some modest reassessment of analyses and methods.

Following the advice of Reviewer 1, we have revised our methods to more consistently categorize herbivore taxa based on physical characters, the most straightforward of which is tooth crown height; this different approach did not substantially change our prior interpretations or conclusions that browsers, which mostly have low-crowned teeth, are more constrained in diet. We subsequently added an assessment on dietary breadth and specialization based on median taxon diet. Additional supporting information regarding plant physiology and the geologic records of climate have been added, all of which supports our assessment of C3 and C4 plants and our determination that $\delta^{13}\text{C}$ is a reliable proxy for grazing vs. browsing in our study system (based on their location in space and time). We addressed concerns brought up by Reviewer 2 of directly comparing enamel sampled from low-crowned and high-crowned taxa by clearing up some misconceptions about how sampling occurs. We also provided data from a separate study (in review) that addresses individual variation in tooth enamel and Reviewer 2's perceived issues surrounding temporal scale and mismatch. The assumptions that Reviewer 2 noted as "implicit" on our part were informed by this separate study.

.....
Referee: 1

Comments to the Author(s)

Pardi and DeSantis have presented a very interesting study that adds significantly to our understanding of the feeding ecology of Neogene North American herbivores. I think the study is of appropriate relevance and broad interest to be considered for publication in the Proceedings of the Royal Society B. However, I have several comments and concerns that I would like the authors to consider. My concerns and comments are detailed below.

We appreciate the positive review and have incorporated the suggested changes, below.

Morphologically-inferred diets

There are inconsistencies in the morphological traits that the authors cite as evidence to morphologically infer the diet of the taxa in their study. Based on the references provided in Table S1, some of the taxa are assigned to a dietary category based on evidence from dental microwear (Capromeryx, Cormohipparion, Stockoceros, and Tetrameryx), degree of hyposonty (13 out of the 30 taxa studied), premaxillary shape (all camelids), changes in enamel arrangement and/or structure (Cuvieronius and Mammuthus), and multivariate assessment of

craniodental features (*Pediomeryx*). In my opinion, the application of dental microwear and degree of hypsodonty to morphologically infer the diet of the taxa in the study are the most problematic and my concerns are expressed below.

The authors assigned the extinct antilocaprids (*Capromeryx*, *Stockoceros*, and *Tetrameryx*) to a dietary category based on the study by Semprebon and Rivals (2007). Semprebon and Rivals (2007) made dietary inferences for those taxa based on their analysis of dental microwear. As Pardi and DeSantis point out in the Introduction of their manuscript, dental microwear and stable isotopes provide insights into “what an individual animal consumed at a specific place and moment in time, as opposed to its potential diet as inferred from morphological features...” (Pardi and DeSantis page 4). Therefore, the authors should reassess the dietary assignment of the extinct antilocaprids using proper morphological traits. For the equids *Cormohipparion*, *Hipparion*, and *Onohippidium* the authors cite MacFadden (2005) as the source of their morphologically assigned diet. MacFadden (2005) presents a figure that summarizes the phylogeny, geographic distribution, diet, and body sizes of the family Equidae at the generic level. In this figure, *Cormohipparion* is identified as a mixed-feeder, *Hipparion* as a grazer, and *Onohippidium* as a mixed-feeder. MacFadden (2005) does not explicitly state what kind of data was used to infer the diet of the different genera shown in the figure. However, in the text he mentions that some hypsodont equids were grazers and others mixed-feeders, citing the study by MacFadden et al. (1999). The study by MacFadden et al. (1999) inferred the diets of six equid taxa (including *Cormohipparion emsliei*) from the Bone Valley deposits of central Florida based on analyses of stable carbon isotopes and dental microwear. Both dental microwear and stable carbon isotopes indicate that *Cormohipparion emsliei* was a mixed-feeder. *Hipparion* and *Onohippidium* are not mentioned in MacFadden et al. (1999) and it is unclear what data MacFadden (2005) used to infer their diets. In any case, the dietary assignments for *Cormohipparion*, *Hipparion*, and *Onohippidium* need to be reassessed using proper morphological traits.

According to Table S1, most of the taxa in the study (13 out of 30) were assigned to a dietary category based on the data reported in Janis et al. (2004). Janis et al. (2004) used relative tooth crown height to separate ungulates with a potentially browsing diet from ungulates with other dietary strategies. All low-crowned (brachydont and submesodont; hypsodonty index ≤ 2.5) taxa in that study were assumed to be browsers or animals with a predominantly browsing diet (Janis et al. 2004). Non-brachydont taxa (hypsodonty index > 2.5 ; mesodont and hypsodont taxa) were assumed to have been potentially grazing and/or mixed feeding. Janis et al. (2004) did not attempt to separate potentially grazing taxa from potentially mixed feeding taxa, but rather referred to those taxa collectively as non-brachydonts. Therefore, it is unclear to me how the authors assigned non-brachydont taxa to grazing and mixed-feeding dietary categories. This should be clarified in the manuscript.

We appreciate the reviewers comment. Some papers we previously cited blur the boundary as they often included morphology and sometimes other proxy data. However, we now ensure that all categorizations for the relevant analyses **only** include consistent morphological characters. Further, see the comments below for the use of tooth crown height, which we have now adopted.

A more important issue is that the morphologically-inferred diets presented by the authors were not obtained in a consistent way. Premaxillary shape and other craniodental traits were used in some taxa, whereas the degree of hypsodonty or changes in enamel structure were used in other taxa. This is a concern because these traits can be discordant. For example, *Antilocapra* and *Megatylopus* show discordant dietary reconstructions when looking at their degree of hypsodonty and premaxillary shape. *Antilocapra* is a highly hypsodont ungulate and *Megatylopus* is a mesodont ungulate (Dompiere and Churcher 1996; Janis et al. 2000; Janis et

al. 2004; Semprebon et al. 2019); therefore, on the basis of this trait alone they could be interpreted as mixed-feeders or grazers. In contrast, Antilocapra and Megatylopus have narrow and pointed premaxillaries, which indicate a browsing or predominantly browsing diet (Dompierre and Churcher 1996). One of the factors that very likely contributes to the discordance between these morphological traits is that tooth crown height is known to correlate not only with diet, but also to the degree of openness of the habitat (Janis 1988; Janis et al. 2002; Damuth and Janis, 2011). Therefore, hypsodonty is not necessarily an adaptation for grazing or mixed-feeding, but rather an adaptation to sustain high levels of dental wear, which can arise as a result of eating abrasive plants (such as grasses) and/or ingesting significant amounts of dust and grit (Janis and Fortelius 1988; Damuth and Janis, 2011). Some researchers have even suggested that the development of hypsodonty was primarily driven by an increased exposure to exogenous grit (Semprebon et al. 2019). Relative to other craniodental traits, hypsodonty has been found to be problematic in determining the diet of some ungulates (e.g., Semprebon et al. 2004). With these considerations in mind, if the authors want to use the degree of hypsodonty as a morphological trait that can be consistently evaluated across all taxa in their study, my recommendation is for them to substitute the notion of morphologically-inferred diets and replace it with categories of hypsodonty (i.e., brachydont, submesodont, mesodont, and hypsodont). This will reveal that all brachydont taxa are “locked” as browsers, except for Platygonus and Mylohyus which extend into the mixed C3-C4 region, possibly as a result of an omnivorous diet. Many hypsodont taxa have $\delta^{13}\text{C}$ values that extend well into the C4 region, but are not precluded from consuming C3 plants. Other hypsodont taxa (e.g., Antilocapra, Capromeryx, and Stockoceros) virtually consume only C3 plants, potentially indicating a uniquely or mostly browsing diet. Some mesodont taxa (e.g., Hemiauchenia) span the full C3-C4 spectrum, whereas other mesodont taxa (e.g., Megatylopus) have only a C3 diet.

Reviewer 1 provides a good and appreciated solution for using morphological traits that can be consistently evaluated across all the taxa in our study, and that is to substitute the notion of “morphologically-inferred diets” with categories of hypsodonty. This categorization is straightforward and consistent within Artiodactyls and Perissodactyls. Damuth and Janis 2011; Janis, Damuth, and Theodor 2004, described degrees of hypsodonty using a “hypsodonty index” (HI) calculated as the lower m3 crown height divided by the lower m3 width. In their categorization, hypsodont taxa possess an HI >3.0 and highly hypsodont taxa possess an HI > 4.5. Other investigators quantify hypsodonty somewhat differently (such as crown height/length (MacFadden and Shockey 1997)). Further, others have described the relative crown height of taxa categorically, rather than as a continuous character. As Damuth and Janis 2011 point out, such differences do add some confusion when comparing indices across studies – but the overall concept of brachydont (low crowned), mesodont (medium crowned), and hypsodont (high crowned) are similar. For our statistical analyses, we considered anything with an HI >3, sensu Janis 1988, as “hypsodont”, but note the diets and trends of highly hypsodont taxa (HI>4.5) for relative visual comparison. We follow reported categorical descriptions of tooth crown height when a numerical HI is not available. Taxa in the Proboscidea were similarly categorized by crown height, but we additionally note characterizations provided by Lambert and Shoshani 1998. Categorizations and descriptions of the relative hypsodonty of the taxa in this study can be found in Supplemental Table S1, along with references, and now replace the concept of “morphologically-inferred diets.” The authors also directly consulted with C. Janis on the final categorizations and made adjustments as necessary. These are also noted in Table S1.

In this revised analysis comparing hypsodonty with diet, we compared the HI categorization with categories defined by the isotope data in our study. Taxa were

categorized as primarily “browsing”, “mixed-feeding”, or “grazing” based on their overall median diet, and taxa were secondarily classified by the breadth of the interquartile range. Taxa with median diets where the third quartile did not exceed -9‰ were classified as “browsers”, while “browser/mixed-feeders” had median values < -9‰, but a third quartile that exceeded this threshold. “Mixed-feeders” were those taxa where the first and third quartiles were between -9‰ and -2‰. “Grazers” were taxa where the third quartile was > -2‰.

We have also revised how we set up our research questions and additionally ask “are grazers specialists?” Morphological categorization is not necessary to evaluate this question, and we assessed whether there was a significant relationship between median taxon diet and taxon breadth.

Stable isotope analysis

The main assumption that the authors make in their study is that $\delta^{13}C$ values can reliably distinguish between animals that were grazing, animals that were browsing, and animals that were mixed feeding. A C3 diet is considered to be indicative of browsing, a C4 diet indicative of grazing, and an isotopically mixed diet indicative of mixed feeding. However, this is an assumption that may not hold true across all localities that the authors studied. The authors indicate that restricting their analysis to localities found at lower latitudes (<37° latitude) mitigates the confounding effects of C3 grasses. However, not only is there a latitudinal gradient in C3-C4 grass abundance, but there is also evidence for a general longitudinal gradient, at least during the late Pleistocene in the American Southwest (Connin et al. 1998). C4 grasses appear to decrease in abundance from east to west (Connin et al. 1998). Other studies also suggest decreased C4 grass abundance in different areas of the American Southwest relative to other geographic regions during the Last Glacial Maximum, the mid-Holocene, and the recent past (Cotton et al. 2016). The interaction between precipitation patterns, temperature, and topography could account for the patterns in C3-C4 grass abundance.

I am not sure how the authors can account for the potential longitudinal gradient in C3-C4 grass abundance, other than by geographically restricting their analysis even more. The authors can explore this possibility.

In the main body of the paper, we present the results collectively; however, the revised study explores the possibility of a longitudinal gradient in C3/C4 grass abundance in the supplemental materials. We divided our data into regions: the Southwest includes records from southern California, Nevada, and Arizona, South-Central includes New Mexico, Texas, and Oklahoma, and Southeast includes the remaining states east of Texas below 37 degrees latitude. This treatment of the data is now reflected in supplementary Tables S2 and S3, where we provide the median of each taxon by region. It is also reflected in supplemental figures S6-S8.

We note that there are small differences between this treatment of the data and the combined analysis, but these are immaterial and do not appreciably change the major conclusions of the study. Importantly, regional assessments of diet do not change our interpretation of the main taxa in our study. We note this on lines 193-194 of the revised manuscript.

At the very least they need to be clear about their assumptions and models. Therefore, my recommendation to the authors is that they clearly layout their assumptions and acknowledge any potential factors that may confound their results and interpretations. The authors can take a look at MacFadden (2008, section 6.1); I think he does a good job of laying out the assumptions and the model he used to interpret the stable carbon isotope values in his study.

Thank you for directing us to a study that does a good job outlining these assumptions. We have revised our manuscript to better reflect our acknowledgment of these limitations. More detailed discussion of the effects of potential confounding factors can now be found beginning at line 263.

It is also important to point out that the study by MacFadden et al. (1999) provides evidence to suggest that some Hemphillian horses from Florida were grazing on both C3 and C4 grasses. MacFadden et al. (1999) conducted analyses of stable carbon isotopes and dental microwear for six species of horses from the Bone Valley deposits of central Florida. Two of these taxa (Pseudhipparion simpsoni and Nannippus minor) had dental microwear patterns typical of grazing animals, whereas their stable carbon isotope values suggested a primarily isotopically mixed diet. The results of these analyses potentially indicate the presence of C3 grasses in central Florida during the late Hemphillian.

We now include the findings of MacFadden et al. 1999 when interpreting the mixed-feeding signal that is present in horses within our study (lines 264-266). What is clear from the regional treatment through time is that *Equus* was consuming a mixture of C3 and C4 resources in the Rancholabrean and Latest Quaternary in the Southeast, where the abundance of C3 grasses should be the least abundant.

On a separate note, I realized that the authors did not include the stable carbon isotope data reported by MacFadden et al. (1999) and MacFadden (2008). The authors should incorporate those data in their study, given that one of the objectives of the paper is to provide a synthesis of the stable isotope data of North American ungulates over the past 7 million years. The studies by MacFadden et al. (1999) and MacFadden (2008) provide stable carbon isotope data for two taxa that are omitted in the manuscript by Pardi and DeSantis (Astrohippus and Nannippus) in addition to data for four taxa that are included in the manuscript.

Omission of these publications was not an oversight. In addition to providing new data, the study of MacFadden 2008 includes an analysis of a subset of data from MacFadden, Solounias, and Cerling 1999. The 2008 paper does not provide raw data values, which precludes their inclusion in calculating the median values and breadth in our study. Unfortunately, the MacFadden et al. 1999 paper also does not present the raw data values in the main body of the publication, and the journal web address that should redirect to additional supplementary materials returns an "Error 404" response. To date, we have been unable to obtain these raw data. Considering these obstacles while acknowledging the reviewer's point about including these publications in a synthesis, we have added the relevant means and standard deviations for the two additional taxa within MacFadden 2008 to supplementary tables S1 and S2.

Other general comments

I think it would be very useful if the authors could add maps indicating the geographic location of the fossil localities for each time interval they studied. This will make it easier for the reader to visualize the geographic and temporal distribution of the fossil sites included in their study. The maps can be included in the Supplementary Information.

We agree that maps indicating the geographic locations of the fossil localities for each time interval would be useful and helpful. To the best of our ability, we have found the locations for the sites included in our study and have provided maps in the supplemental materials.

.....

Referee: 2

Comments to the Author(s)

The present manuscript is an analysis of the congruence between morphological and stable isotopic metrics for paleodiet. The authors compile dietary classifications based on morphology from the literature as well as published stable carbon isotope values from carbonates. Briefly, they find that morphology and reconstructed diet based on $\delta^{13}\text{C}$ values are not always congruent. In particular, species reconstructed as browsers using morphology are more commonly reconstructed as C3 browsers using isotopes than species classified as grazers are reconstructed as C4 grazers. They also show that species at individual sites tend to show lower dietary variation than that of the entire species to which they belong. Furthermore, the dietary breadths of species at single localities are not correlated with estimates of plant abundance, suggesting that competition and site-specific specialization may be at play. The authors suggest that the evolution of grazing "adaptations" such as hypsodonty enable ungulate species to take advantage of a larger range of food stuffs while browsers, by virtue of their brachydont tooth condition, are more dietarily limited. The authors therefore recommend a revision of the categorical terms used to describe dietary categorizations based on morphology.

Overall, the manuscript is well written and the arguments are easy to follow. There are, however, some methodological issues, as outlined below.

The first is that there is a fundamental mismatch between the temporal scales at which morphology and stable isotopes are used to define diet. Changes in morphology represent hundreds to millions of years of evolution. In contrast, stable isotopes represent days to years of an animal's life. In the case of a single serial sample from a molar, the $\delta^{13}\text{C}$ values could represent a few days of life (depending on mineralization and maturation rates) while, depending on the taxon, a bulk sample could represent months to years. As a result, we do not necessarily expect 100% congruence of diet reconstructions from morphology and stable isotopes.

We agree, and clarify this point in the introduction (Lines 64-66) however, we also note here that morphology is frequently used to make ecological assessments of fossil communities. The objective of our study is to identify and quantify the mismatch between the inferred ecological significance of taxa based on their morphology vs. their actual behavior.

In the context of the present manuscript, however, the authors seem to implicitly assume that reconstructions from stable carbon isotopes are representative of lifetime diet, which they demonstrably are not. There is explicit recognition in the stable isotope literature of short term variations in diet that could explain some of the patterns reported in the present manuscript.

Thank you for the opportunity to clarify some assumptions of our study. The authors have a separate publication that was in review that has informed our understanding of individual specialization (a copy of that publication has been provided to the editorial staff). Our revisions now include a subset of that publication's data that can address the reviewer's concerns about short term variations in diet. Briefly, that study uses 4134 serial samples of tooth enamel from 318 individuals from around the globe, and finds that the vast majority (89%, N=283 of 318) range $\leq 3\%$ in $\delta^{13}\text{C}$, across feeding strategies (browsing, mixed-feeding, grazing). When those analyses were restricted to individuals that were hypsodont enough to allow for at least 5 serial samples, we still find that 87.9%

(246 of 280) have isotopic diets that range less than 3‰ across serial samples. Globally, no browsers have ranges exceeding 3.3‰, no mixed-feeders range more than 3.9‰, and a small percentage of grazers (14%) had ranges greater than 3‰. Below 37 degrees latitude (the geographic restriction we have applied in the present study), 82.6% (138 out of 167) of herbivores have individual ranges \leq 3‰. To summarize: despite observed variations in diet, most individuals within herbivorous taxa consume relatively stable diets, ranging less than a few per mil, over extended periods of time (months to years). For the present study, we have now provided a similar analysis within the supplemental information for a subset of our data for which there were serially sampled isotopes. We find the same overall result: most individuals do not have a maximum range in their recorded diet of more than a few per mil. Within that subset (N=92), 78.3% (N=72) have ranges that are \leq 3‰. Just among individuals from hypsodont taxa (N=66), 69.7% (N=46) have ranges \leq 3‰.

Morphological and stable isotopic methods are testing different hypotheses relating to diet.

The objective of our study is to identify and quantify the mismatch between the inferred ecological significance of taxa based on their morphology vs. their actual behavior. Much work has been done correlating morphology with diets (Janis 1990), and morphology has often been used to infer the paleoecology of faunas, especially studies that pre-date individual-level dietary proxies. **Given this historical fact, reviewer 2 presents a compelling argument for why our study is necessary.** Additionally, our study increases our understanding of intraspecific variation in diet within taxa, which has received less attention (MacFadden 2008).

*Consideration of the differences in temporal scale are important because this is why people see a deer (*Odocoileus*) eat a dead bird and conclude that they are omnivores or see a deer eat grass and conclude they are grazers, when they are in fact, on average, browsers. To this end, mixing of brachydont and hypsodont taxa, whose teeth record highly variable amounts of time, could incidentally produce the entire result presented in the manuscript. A single molar of a horse or bison represents more than a year of bulk diet. According to Hoppe et al. some individual horse teeth may record diet over 2.8 years of the animal's life. In contrast, the teeth of brachydont species (deer, for example) mineralize and mature over < 1 year of the animal's life. In this case, the authors are comparing apples and oranges. The observation of greater dietary specialization for browsers is exactly what one might expect simply due to a temporal mismatch. As a result, I see no real way to differentiate dietary specialization from the simple fact that teeth from browser-adapted and grazer-adapted species represent different amounts of the animal's lifetime and thus different temporal scales over which their diets are recorded.*

Thank you for the opportunity to note the differences between the amount of enamel typically sampled for serial vs. bulk samples. With few exceptions, a bulk sample typically does not sample > ~1cm of enamel parallel to the growth axis of the tooth regardless of whether a taxon is hypsodont or brachydont. This is now noted on line 94. Thus, the overwhelming majority of bulk samples allow for an apples-to-apples comparison in the amount of enamel compared. We acknowledge that serially sampling a hypsodont tooth can result in a greater amount of time represented, and provide a version of Fig. 1 from the main manuscript in the supplemental materials, Fig. S7, where we only consider bulk data. We note that there is no appreciable difference, which is unsurprising given that bulk samples make up the majority of the data in our study and, our study in review finds that variation within an individual tooth is typically very low.

Furthermore, C3 plants have more variable stable carbon isotope ratios. According to Cerling et al. the standard deviation for C3 plants is ~2.7 ‰ whereas for C4 plants it is 1.1‰.

The narrow standard deviation of C4 plants, and the lack of overlap in $\delta^{13}\text{C}$ of enamel sampled from C3 and C4 specialists in Cerling et al. 1997 would suggest that a taxon would necessarily need to consume a mixed diet, or selectively consume rare C3-grass resources, to have a median $\delta^{13}\text{C}$ value much less than -2‰. The relative percentage of C3 and C4 plants can be estimated from a mixing model (MacFadden and Cerling 1996): assuming a mean $\delta^{13}\text{C}$ value for C3 plants of -27‰ and a mean value for C4 plants of -13‰, the overall lower quartile observed for *Equus* of -5.4‰ would be obtained with a diet that is ~40% C3 vegetation. If we consider only sites from the SE, where the abundance of C3 grasses would be at their lowest (Paruelo and Lauenroth 1996; Teeri and Stowe 1976), a lower quartile range of -4.1 would be obtained with a diet composed of more than 30% C3 vegetation. While it is true that somewhat less enriched $\delta^{13}\text{C}$ values among grazers can be attributed to the consumption of C3-grasses, this becomes less likely for animals with more depleted values that are occurring in C4-grass dominated systems.

So, ability to make comparisons of similar temporal scale for brachydont and hypsodont taxa might result in findings different from those presented in the manuscript. This could be done by looking at comparable amounts of enamel for brachydont species and hypsodont species for which they possess data from serial sampling.

We acknowledge that brachydont and hypsodont teeth may record highly variable amounts of time, and we agree with the reviewer that looking at comparable amounts of enamel among brachydont and hypsodont taxa will allow us to make more comparable dietary determinations. Additionally, the location where serial samples are taken along a hypsodont tooth is potentially something that needs to be considered when comparing with a brachydont taxon. We anticipated this and considered this in our serial sample study, in review. To account for differences in sample location along hypsodont teeth, stats were calculated using a five-sample sliding window. For each individual, the range in $\delta^{13}\text{C}$ was calculated across a sliding window and the average range computed. This treatment helped account for potential differences in the number of serial samples taken from different kinds of teeth (fewer from brachydont vs. many from hypsodont), which as the reviewer noted, may result in comparing different amounts of time. This treatment of the data explored whether drawing samples from different locations along the growth axis of a hypsodont tooth results in different dietary interpretations – it does not. It also helps account for any variation in the calculated range based on where samples are taken along the growth axis of a more hypsodont tooth in comparison to a brachydont tooth. Using the sliding window approach, we found the same right-skewed result as from the raw data: 97% of individuals had ranges $\leq 3\text{‰}$.

A couple of other important points are missed by the authors. The first is that hypsodont taxa (particularly horses because they are hindgut fermenters) require that their teeth be worn down by the consumption of abrasives (whether this comes from grass or exogenous abrasives is a point of debate, although people seem to have settled on the latter). If not worn down, the teeth will not wear evenly and individuals will be unable to eat (a condition referred to as wave mouth). This is precisely why veterinarians (and farriers) file horse teeth. As such, horses cannot subsist for long periods of time on non-abrasive foods. Thus, the above mentioned mismatch in temporal scale becomes even more important to consider.

We address perceived issues of temporal mismatch in our responses above

In the same vein, the paper lacks any discussion of the contribution of abrasives. Perhaps consumption of exogenous abrasives by grazing-adapted species ameliorated filled the tooth wear requirement.

This is an interesting idea. Attempting to quantify the minimum amount of grit required by hyposodont taxa would be, to our knowledge, an unexplored aspect of the niche of herbivores at the scale we are addressing. Although outside the scope of our study, we mention this interesting hypothesis briefly in the discussion of our revised manuscript (line 337-342).

The authors have attempted to exclude the possibility of C3 grazing by limiting their samples to below 37 degrees of latitude based on the fact that a predictable gradient in C3/C4 grass abundance is present in modern day North America. They, however, have not presented any evidence to corroborate their assumption that the same was true during the latest Miocene and Pliocene. In fact, they present no climate reconstruction or similar that might strengthen their argument. The work of David Fox on stable carbon isotopes from pedogenic carbonates suggests that the modern C3/C4 latitudinal gradient evolved as late as the Pliocene (some of the sites do fall near or right on 37 degrees of latitude, depending on the map projection). In the below cited paper, Fox et al. suggest that the first appearance of ecosystems having >70% C4 plant biomass is observed only 1.3 Ma. But, the assumption that 37 degrees represents a good cut off and a means to reject C3 grazing may, in fact, not be accurate for part of the time period covered by the present study. Even today, ecosystems below 37 degrees of latitude can be between 70 and 100% C4.

In line 262 of the manuscript we direct readers to the supplemental materials, in which we detail the energetic tradeoffs that result in the competitive exclusion of C3 grasses by C4 grasses under low atmospheric CO₂ levels in warm environments (Ehleringer, Cerling, and Helliker 1997; Ehleringer and Pearcy 1983). We cite evidence of low atmospheric CO₂ for the last 7 million years (Gerhart and Ward 2010; Tripathi, Roberts, and Eagle 2009), and the rationale for a consistent gradient in C3/C4 grass abundance (Cerling, Ehleringer, and Harris 1998; Strömberg and McInerney 2011). We would also like to note that although we gathered data from the literature documenting SIA of herbivore diets since ~7 Ma, these data are largely comprised of Pleistocene records. There is also spatial bias in the body of literature of low latitude mammalian SIA diets, with the majority coming from eastern localities, where the abundance of C₃ grasses should be at their lowest. We have broken out our characterizations of diets by region in the Supplemental materials, and note that there is very little regional variation in overall taxon dietary interpretations from SIA. We thank the reviewer for directing us to the study by David Fox (Fox et al. 2012), and note that although their study was conducted in the “southern Great Plains”, the geographic extent of their study (Kansas) was outside of the geographic constraints of our study. The Fox study also notes that “The somewhat lower mean $\delta^{13}\text{C}$ value and low abundance of C₄ biomass during the Clarendonian in the Meade area, and perhaps western Kansas in general, appear to represent local variation in comparison to the Great Plains as a whole.”

The authors use a metric for the availability of C3 and C4 resources (the minimum for browsers and maximum for grazers) but do not provide a citation. Does this work in assemblages of mammals where the composition of the ecosystem is known? In mixed, mid-latitude C3/C4

ecosystems do grazers always show a mixed C3/C4 signal? I could see this being true, given that grazers don't necessarily differentiate between C3 and C4 grasses. But it could also be untrue if there is spatial segregation of species with different diets (e.g. grazers feeding in warmer, open areas with more C4 as opposed to the cool understory with more C3). I primarily want to see more justification for their metric. Perhaps this could come in the form of paleoecological reconstructions of the associated plant ecosystems.

We appreciate this comment and after further consideration, we reconsidered using community-level characterizations of sites where we could not be certain that the ecological community was isotopically characterized. After reframing some of our research questions, this analysis became tangential to the objectives of the main text, so this analysis was removed.

There is an implicit assumption that if individual animals are found at sites below 37 degrees of latitude that they never migrated etc. to more northern parts of the continental USA. Some modern equids and ruminants do undergo relatively long-distance seasonal movements. There is no reason to assume that Miocene-Pleistocene ungulates did not. Moreover, Bison are included in their sample and are well known for their north-south migrations in North America. So, though the authors interpret a mixed C3/C4 isotopic signal as representing a mixed browsing/grazing signal, I do not believe they have the power to reject C3 grazing. The associated oxygen isotope data may, however, provide some illumination, though the authors would have to work to differentiate seasonal and migration effects.

We initially shared this same concern with migration, however our study in review of serially sampled individuals does not support large fluctuations between C3 and C4 diets. That study also finds no significant relationship between body size (which is significantly related to home range) and the range in $\delta^{13}\text{C}$ values observed across individuals' serial samples. We address issues of migration in the discussion of our manuscript beginning on line 263.

The authors test for differences in the range of isotopic values between individuals at local sites to the entire range of variation for the species (within a time bin, I believe). There is a mathematical issue with this approach because they have not, to my understanding of the paper, demonstrated that localized patterns are differentiable from a random draw from the entire "population" or set of samples. In fact, they clearly demonstrate what I would expect simply from random sampling that variation at the entire sample scale is higher than at the local scale. Differentiating random effects from localized specialization could be achieved by using something like Cohen's D where observed values are compared to a set of randomizations. If, local values fall outside the range of the randomized values, the authors would have a stronger argument for localized dietary specialization.

Thank you for suggesting that we use a standardized mean effect size to differentiate random effects from localized specialization, which is now included in the methods. Briefly, we randomized samples taken from occurrences within each NALMA to find the expected local breadth if local assemblages were a random sample of the available taxon pool. We compared the mean range calculated from these randomizations to the observed mean ranges using Cohen's D, and assigned magnitudes of effect size (Sawilowsky 2009). Details of this analysis can be found in the supplemental materials. This comparison of expected to observed local breadth presents a different, but complementary, approach to our original presentation of average local breadth as a fraction of a taxon's overall breadth. Our initial analysis tested whether the average local

fraction represented is different across taxa, while this new analysis further indicates whether a taxon's local breadth is narrower, or broader, than would be expected by chance. The latter analysis does a better job of assessing specialization.

A minor point of contention involves the author's statement that stable isotope values are phylogenetically independent (without relevant citation) (line 65). This is demonstrably untrue. At the very least, I would expect a division between the artiodactyls and perissodactyls based on their dietary physiology, which results in different degrees of carbon and nitrogen fractionation. I think it is naïve to assume that reconstructions of diet from stable isotopes are "taxon free." This requires the assumption that physiology is invariant across mammalian taxa. We know this very likely isn't true. However, I reiterate that this is a minor point because their analyses probably do not require phylogenetic correction.

Since we agree that no phylogenetic correction is needed, we have simply removed the unnecessary reference to phylogenetic independence or any mention of "taxon free".

In Summary: We are very appreciative of the detailed comments provided by the reviewers and have made significant revisions to this paper which reflect these comments and serve to strengthen the paper. Subsequently, the paper now has 4 Figures, 2 Tables, 4 Supplemental Tables, and 9 Supplemental Figures which reflect these revisions. Collectively, this paper helps demonstrate the degrees to which ecological interpretations made through morphology align with SIA, and increases our understanding of dietary variation within taxa. Our study is of broad interest to paleontologists, evolutionary biologists, ecologists, and even conservationists and an ideal fit for Proceedings of the Royal Society B. Thank you for your continued consideration of this manuscript.

References Cited:

- Cerling, T. E., J. R. Ehleringer, and J. M. Harris. 1998. "Carbon Dioxide Starvation, the Development of C4 Ecosystems, and Mammalian Evolution." *Philosophical Transactions of the Royal Society B: Biological Sciences* 353(1365):159–71.
- Damuth, John, and Christine M. Janis. 2011. "On the Relationship between Hypsodonty and Feeding Ecology in Ungulate Mammals, and Its Utility in Palaeoecology." *Biological Reviews* 26.
- Ehleringer, James, and Robert W. Pearcy. 1983. "Variation in Quantum Yield for CO₂ Uptake among C₃ and C₄ Plants." *Plant Physiology* 73(3):555–59. doi: 10.1104/pp.73.3.555.
- Ehleringer, James R., Thure E. Cerling, and Brent R. Helliker. 1997. "C₄ Photosynthesis, Atmospheric CO₂, and Climate." *Oecologia* 112(3):285–99. doi: 10.1007/s004420050311.
- Fox, David L., James G. Honey, Robert A. Martin, and Pablo Peláez-Campomanes. 2012. "Pedogenic Carbonate Stable Isotope Record of Environmental Change during the Neogene in the Southern Great Plains, Southwest Kansas, USA: Carbon Isotopes and the Evolution of C₄-Dominated Grasslands." *GSA Bulletin* 124(3–4):444–62. doi: 10.1130/B30401.1.
- Gerhart, Laci M., and Joy K. Ward. 2010. "Plant Responses to Low [CO₂] of the Past." *New Phytologist* 188(3):674–95. doi: <https://doi.org/10.1111/j.1469-8137.2010.03441.x>.

- Janis, Christine M. 1988. "An Estimation of Tooth Volume and Hypsodonty Indices in Ungulate Mammals, and the Correlation of These Factors with Dietary Preferences." *Mémoires Du Museum National d'Histoire Naturelle* 53:367–87.
- Janis, Christine M. 1990. "Correlation of Cranial and Dental Variables with Body Size in Ungulates and Macropodoids." Pp. 255–300 in *Body Size in Mammalian Paleobiology: Estimation and Biological Implications*, edited by J. D. Damuth and B. J. MacFadden. Cambridge: Cambridge University Press.
- Janis, Christine M., John Damuth, and Jessica M. Theodor. 2004. "The Species Richness of Miocene Browsers, and Implications for Habitat Type and Primary Productivity in the North American Grassland Biome." *Palaeogeography, Palaeoclimatology, Palaeoecology* 207(3):371–98. doi: 10.1016/j.palaeo.2003.09.032.
- Lambert, D. W., and J. Shoshani. 1998. "Proboscidea." Pp. 606–21 in *Evolution of Tertiary Mammals of North America*. Vol. 1, edited by C. M. Janis, K. M. Scott, and L. L. Jacobs.
- MacFadden, Bruce J. 2008. "Geographic Variation in Diets of Ancient Populations of 5-Million-Year-Old (Early Pliocene) Horses from Southern North America." *Palaeogeography, Palaeoclimatology, Palaeoecology* 266(1):83–94. doi: 10.1016/j.palaeo.2008.03.019.
- MacFadden, Bruce J., and Thure E. Cerling. 1996. "Mammalian Herbivore Communities, Ancient Feeding Ecology, and Carbon Isotopes: A 10 Million-Year Sequence from the Neogene of Florida." *Journal of Vertebrate Paleontology* 16(1):103–15.
- MacFadden, Bruce J., and Bruce J. Shockey. 1997. "Ancient Feeding Ecology and Niche Differentiation of Pleistocene Mammalian Herbivores from Tarija, Bolivia: Morphological and Isotopic Evidence." *Paleobiology* 23(1):77–100.
- MacFadden, Bruce J., Nikos Solounias, and Thure E. Cerling. 1999. "Ancient Diets, Ecology, and Extinction of 5-Million-Year-Old Horses from Florida." 283:5.
- Paruelo, Jose M., and W. K. Lauenroth. 1996. "Relative Abundance of Plant Functional Types in Grasslands and Shrublands of North America." *Ecological Applications* 6(4):1212–24. doi: 10.2307/2269602.
- Sawilowsky, Shlomo S. 2009. "New Effect Size Rules of Thumb." *Journal of Modern Applied Statistical Methods* 8(2):597–99. doi: 10.22237/jmasm/1257035100.
- Strömberg, Caroline A. E., and Francesca A. McInerney. 2011. "The Neogene Transition from C3 to C4 Grasslands in North America: Assemblage Analysis of Fossil Phytoliths." *Paleobiology* 37(1):50–71. doi: 10.1666/09067.1.
- Teeri, J. A., and L. G. Stowe. 1976. "Climatic Patterns and the Distribution of C₄ Grasses in North America." *Oecologia* 23(1):1–12.
- Tripathi, A. K., C. D. Roberts, and R. A. Eagle. 2009. "Coupling of CO₂ and Ice Sheet Stability Over Major Climate Transitions of the Last 20 Million Years." *Science* 326(5958):1394–97. doi: 10.1126/science.1178296.

Appendix B

I reviewed a previous iteration of this manuscript. The manuscript remains well written and, potentially, of broad interest to palaeobiologists. They have addressed many of my concerns (particularly regarding standardized effect sizes and the timescales of morphology vs. stable isotopes). I am mostly satisfied with their responses to these comments and the changes made to the manuscript (I will make some small suggestions for areas where the clarity of the writing can be approved; there are a few spots that confused me, potentially due to the need for more coffee).

The paper is an interesting contribution to the palaeoecological literature in the sense that it addresses the degree of dietary variability at the locality and more regional scales. This is a contribution to the literature that I would like to see published.

However, I do not believe they have adequately addressed my primary concern from the previous review, which is that, during the Miocene and even today, ecosystems below 37 degrees north are not 100% C4. I am aware of the physiological reasons for C4 dominance at lower latitudes in North America, but the fact remains that they are not 100% dominant. This, as I suggested previously, muddies the waters re: their assumption that mixed C3/C4 signals from stable isotopes represent the inclusion of browse in the diet rather than grazing on both C3 and C4 grasses.

Firstly, I clearly directed the authors to the wrong Fox paper. This is the one I meant to reference. I apologize for the somewhat inappropriate reference I included before. This one is much more germane.

Fox, David L., and Paul L. Koch. "Carbon and oxygen isotopic variability in Neogene paleosol carbonates: constraints on the evolution of the C4-grasslands of the Great Plains, USA." *Palaeogeography, Palaeoclimatology, Palaeoecology* 207.3-4 (2004): 305-329.

Figure 4 clearly shows that below 37 degrees north, %C4 never rises above ~40% during the Miocene. During the Plio-Pleistocene, they report some %C4 values within the modern range. Of course, Fox and Koch (2004) are interested in the Great Plains and the present study is limited primarily to Florida and California.

Fig. 1. Map of localities sampled in this study. Some localities are too close together to be distinguished at this scale, hence only 20 symbols are visible.

Fig. 4. Geographic variation in Great Plains paleosol carbonate $\delta^{13}\text{C}$ values and estimated percentage of C_4 biomass during the Neogene. ●, Miocene samples. ○, Plio-Pleistocene samples from southwestern Kansas. (A) Latitudinal gradient in $\delta^{13}\text{C}$ values and percent C_4 biomass. (B) Longitudinal gradient in $\delta^{13}\text{C}$ values and percent C_4 biomass.

But we need only to then look to modern distributions of C_4 plants to show that the assumption based on 37 degrees north does not work.

Osborne, Colin P., et al. "A global database of C_4 photosynthesis in grasses." *New Phytologist* 204.3 (2014): 441-446.

Figure 1 shows the global distribution of C_4 grasses (percentage of total grasses).

In Florida and Southern California, which are the areas sampled by the present study, the percentage of C_4 grasses ranges from somewhere between 60% to 70 or 75%. This means that 25-40% of those ecosystems are composed of C_3 grasses. This says nothing of other types of C_3 plants. It is still a sizeable percentage of the ecosystem.

Figure 1

Open in figure viewer | ↓ PowerPoint

Global map of C₄ grass species distributions. (a) Percentage of grass species within each mapping unit that uses the C₄ pathway; (b) the species richness of C₄ grasses in each mapping unit. The map shows species distributions at the Taxonomic Databases Working Group (TDWG) level 3 'botanical country' scale, a biodiversity information standard corresponding largely to political countries, but with large countries subdivided into smaller mapping units (Brummitt *et al.*, 2001).

In the present manuscript, the authors suggest that C3 grazing could only come about (in the regions they sample) by non-optimal foraging strategies. Can this really be true when C3 plants comprise ~ 30% of the ecosystem? If this is the case, it needs to be supported with literature.

Furthermore, the below study interprets the same environments in California very differently (as C3 dominated) from what is assumed by the present manuscript.

Trayler, Robin B., et al. "Inland California during the Pleistocene—Megafaunal stable isotope records reveal new paleoecological and paleoenvironmental insights." *Palaeogeography, Palaeoclimatology, Palaeoecology* 437 (2015): 132-140.

To address my concern, the authors would need to show that, during their sampling period and in their sampling region, %C4 was higher than modern or show that in modern ecosystems grazers living in ecosystems with a similar balance of C3 and C4 grasses always prefer C4 grasses (either by selectivity or habitat preference). Otherwise, I do not believe they can entirely reject C3/C4 grazing.

In my opinion, the component of this study attempting to compare hypsodonty and stable isotopes would be better performed using modern taxa with known diets based on stomach contents or observations. In this way, the authors could actually tease apart instances of grazing on C3 and C4 grasses from C4 grazing mixed with C3 browsing. I admit this would be a much more difficult study, but there are considerable data for African ungulates.

Specific comments from the text (nitpicky things):

Line 54-55 – Not all characteristics of the feeding apparatus are interpreted as the result of direct selection

Fraser, Danielle, and Natalia Rybczynski. "Complexity of ruminant masticatory evolution." *Journal of Morphology* 275.10 (2014): 1093-1102.

Line 143 – a switch in terminology here to “adaptable taxa”

167-169 – a bit of a long sentence that is difficult to read

225-227 – Same comment

Appendix C

Thank you for the opportunity to address the issues brought forth by the reviewer who assessed our submission. Reviewer or Board Member comments are provided in *italics* with author comments to editorial staff and reviewers provided in plain text. A revised version of our submission with tracked changes can be found immediately following our responses.

Associate Editor

Board Member

Comments to Author:

I am recommending acceptance with minor revisions from a single Reviewer, who has provided a short list of editorial revisions to the manuscript and has also recommended Acceptance with Minor Revisions. I find the manuscript to have addressed, through two rounds of reviews, the requested revisions of two reviewers.

We are pleased to hear that the editorial staff feel we have addressed the major revisions needed from the first round of reviewer comments. In this second set of comments, Reviewer 1 provided a thorough check of our manuscript and identified some additional minor issues, which we have addressed. Reviewer 1 identified some geologic ages that were assigned in error to a few occurrences in our data, which we have corrected. We were also able to obtain data for one of the taxa Reviewer 1 had suggested we add in the first set of revision. These corrections and additions do not change the results of any of our analyses, but are now reflected in the figures and supplemental data. They also identified a few typos and sentences that needed clarification. We have fixed all the identified typos and added clarification where needed.

.....
Reviewer(s)' Comments to Author:

Referee: 1

Comments to the Author(s).

The authors have addressed all of the concerns I indicated on my previous review. However, I still have some minor observations and comments that I would like the authors to consider.

We would like to thank this reviewer for their constructive feedback and for the care with which they read our submission. We are pleased that we were able to satisfy the issues they identified in their initial review. We have addressed all of the notes they have provided us in their second review.

I noticed that on Figure 2 and Figure S8, you have some mammoth specimens identified as Blancan in age. Currently, the Irvingtonian NALMA is defined by the first appearance of Mammuthus in North America south of 55o N (Bell et al. 2004). Therefore, by definition, the mammoth specimens you are plotting here cannot be Blancan in age. Please check the age of these specimens and other specimens from that site to make sure they are assigned to the correct time bin. I am sorry I missed this observation on my previous review. The fact that you are now plotting the data by degree of hypsodonty made this error really stick out; Mammuthus is the only hypsodont proboscidean in the dataset.

The observation mentioned above made me look at your raw data in more detail. I noticed that specimens from Inglis 1A are assigned to the Irvingtonian, but I believe this site is currently considered latest Blancan in age (Bell et al. 2004). This and the error mentioned above are the only errors I was able to identify, but I am not completely familiar with all of the localities you studied. So, please make sure that you are being consistent when assigning specimens to each time bin.

Bell, C. J., Lundelius, E. L. Jr., Barnosky, A. D., Graham, R. W., Lindsay, E. H., Ruez, D. R. Jr., et al. 2004. "The Blancan, Irvingtonian, and Rancholabrean Mammal Ages," in Late Cretaceous and Cenozoic Mammals of North America: Biostratigraphy and Geochronology, ed M. O. Woodburne (New York, NY: Columbia University Press), 232–314. doi: 10.7312/wood13040-009

Thank you for taking a closer look at the data and identifying this issue. We have revised the age for Inglis 1A as Blancan. The *Mammuthus* that were marked as *Blancan* came from a site with a Blancan/Rancholabrean mixed-age fauna, Santa Fe River 1, and the original publication of the data had listed those specimens as Blancan. We reached out to the collections manager caring for these specimens and received clarification on how the site is mixed, and he confirmed that the *Mammuthus* were Rancholabrean in age. We were also able to obtain data for one of the taxa you had suggested we add in the first set of revision, *Nannipus*. These corrections and additions do not change the results of any of our analyses, but are now reflected in the figures and supplemental data. We did go through the entire dataset to look for additional errors and did not find any further issues.

Page 4, line 53: There is a typo here. Please change "...evolutions..." to "...evolution's..."

We have made this revision.

Page 4, line 53: To improve clarity, I suggest rephrasing "...examples of an animal's ability to adapt..." to "...examples of the ability of animals to adapt..."

We have accepted the suggested rephrasing and agree it improves the clarity of the writing.

Page 4, line 57: I find this a bit confusing. You state that "...it is also evident that certain forms do not necessitate specific diets...". By forms do you mean morphology, individual animals, or species?

By forms we mean morphology, we have changed the sentence to make this explicitly stated.

Page 4, line 58: Just to be more precise, I suggest changing "...this allows for the inclusion of grass into their diet..." to "...this allows for the inclusion of grass and other abrasive food items into their diet..."

We have accepted the suggested addition.

Page 4, line 65: There is a typo here. Please change "...shaped evolutionary timescales..." to "...shaped in evolutionary timescales..."

We have corrected this typo.

Page 5, lines 72 - 73: Here you ask: "...Are grazing taxa specialists, or is grazing a means to broaden the dietary niche?" For "grazing taxa" are you referring to taxa identified as "grazers" based on morphological criteria or to taxa identified as grazers based on the isotope data, according to the criteria you outline in the methodology? If you are referring to taxa identified as "grazers" based on morphological criteria, I suggest substituting "grazing taxa" for hypsodont taxa: Are hypsodont taxa specialists, or is hypsodonty a means to broaden the dietary niche.

The reviewers concern here is well taken. We were referring to taxa that can graze on the basis of their actual diets, not necessarily taxa that are hypsodont. Given the framing of this paper, however, we have changed the text to emphasize hypsodonty – “Are hypsodont or grazing-adapted taxa specialists, or is hypsodonty a means to broaden the dietary niche?”

Page 7, line 125: There is a typo here. Please change "...were where..." to "...where..."

We have corrected this typo.

Page 10, line 193: There is a typo here. Please change "...Perisodactyls..." to "...Perissodactyls..."

We have corrected this typo.

Page 11, line 213: Please substitute underlined text with normal text.

We have removed the underline.

Page 17, line 359: Do you mean dietary categorizations based on morphology?

We were referring to revising what we call taxa based on their realized diets. The preceding sentence is about the apparent ability of mixed-feeding-adapted and grazing-adapted taxa to consume broad diets across the C3-C4 spectrum; we have added a transition in the final sentence of the paragraph to link them better.

Page 18, line 381: Please change "...grazing megafauna..." to "...grazing-adapted megafauna..."

We have made this change.

Page 18, line 389: I suggest replacing "browsers" with "brachydont taxa", so that you refer to a morphological trait typically associated with browsing.

We have made this change.

Page 29, line 601: Please change "...the latest record specimens..." to "...the oldest recorded specimens..."

We have made this change.

In Summary: We are very appreciative of the detailed comments provided by the reviewer and have made the requested revisions to this paper. Collectively, this paper helps demonstrate the degrees to which ecological interpretations made through morphology align with SIA, and increases our understanding of dietary variation within taxa. Thank you for your consideration, and acceptance, of this manuscript. A copy of the manuscript with tracked-changes follows.